# A-TPT: Angular Diversity Calibration Properties for Test-Time Prompt Tuning of Vision-Language Models

**Shihab Aaqil Ahamed**[1,2]     **Udaya S.K.P. Miriya Thanthrige**[1]     **Ranga Rodrigo**[1]
**Muhammad Haris Khan**[2]
[1]Dept. of Electronic and Telecommunication Engineering, University of Moratuwa, Sri Lanka
[2]Mohamed bin Zayed University of Artificial Intelligence, Abu Dhabi, UAE
shihabaaqilahamed@gmail.com
**Project Page:** https://mb-shihab-aaqil-ahamed.github.io/A-TPT/

## Abstract

Test-time prompt tuning (TPT) has emerged as a promising technique for adapting large vision-language models (VLMs) to unseen tasks without relying on labeled data. However, the lack of dispersion between textual features can hurt calibration performance, which raises concerns about VLMs' reliability, trustworthiness, and safety. Current TPT approaches primarily focus on improving prompt calibration by either maximizing average textual feature dispersion or enforcing orthogonality constraints to encourage angular separation. However, these methods may not always have optimal angular separation between class-wise textual features, which implies overlooking the critical role of angular diversity. To address this, we propose **A-TPT**, a novel TPT framework that introduces angular diversity to encourage uniformity in the distribution of normalized textual features induced by corresponding learnable prompts. This uniformity is achieved by maximizing the minimum pairwise angular distance between features on the unit hypersphere. We show that our approach consistently surpasses state-of-the-art TPT methods in reducing the aggregate average calibration error while maintaining comparable accuracy through extensive experiments with various backbones on different datasets. Notably, our approach exhibits superior zero-shot calibration performance on natural distribution shifts and generalizes well to medical datasets. We provide extensive analyses, including theoretical aspects, to establish the grounding of A-TPT. These results highlight the potency of promoting angular diversity to achieve well-dispersed textual features, significantly improving VLM calibration during test-time adaptation. Our code is available at https://github.com/MB-Shihab-Aaqil-Ahamed/A-TPT/.

## 1 Introduction

Foundational large-scale vision-language models (VLMs), such as CLIP (Radford et al., 2021), ALIGN (Jia et al., 2021), and FILIP (Yao et al., 2021), have demonstrated remarkable zero-shot inference capabilities in a wide range of downstream tasks (Jia et al., 2021; Radford et al., 2021). These models are pre-trained with contrastive learning on massive web-scale data — e.g., 400 million image-text caption pairs — to align visual and textual modalities within a shared multimodal latent space. This alignment allows VLMs to classify instances from novel visual categories in a zero-shot setting realized by carefully constructed textual prompts — hand–crafted class-conditioned templates (e.g., "a photo of a [CLS]") — crucial for effective zero-shot transfer. However, manually designing such prompts often requires domain-specific heuristics and may not be optimal across diverse tasks (Shu et al., 2022).

To address these limitations, recent works have explored prompt tuning that learns prompts from training data specific to downstream tasks (Zhou et al., 2022a;b). However, such approaches often rely on annotated data, which can be expensive and scarce for zero-shot scenarios (Socher et al., 2013). To address this challenge, test-time prompt tuning (TPT) (Shu et al., 2022) has garnered

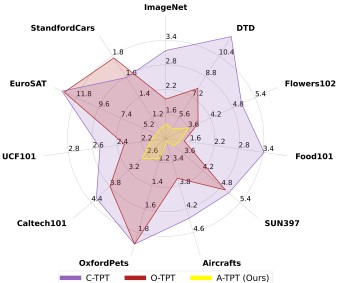

Figure 1: Comparison of calibration performance (ECE) with C-TPT (Yoon et al., 2024), and O-TPT (Sharifdeen et al., 2025) on fine-grained classification datasets with CLIP ViT-B/16 backbone. Ours (lower ECE) shows improved prompt calibration.

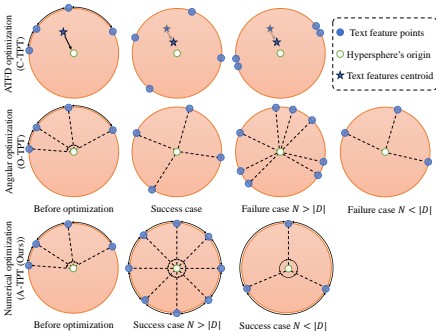

Figure 2: Comparison of numerical optimization (A-TPT (Ours)) with angular optimization (O-TPT Sharifdeen et al. (2025)) and ATFD optimization (C-TPT Yoon et al. (2024)).

significant attention focused on prompt tuning. TPT optimizes learnable prompt vectors through gradient descent that adaptively refines them during inference with unlabelled test image samples to adapt VLMs to novel tasks. Although TPT can boost the accuracy of VLMs, it often suffers from poor calibration, where the model's predicted confidence does not reliably reflect the true accuracy (Guo et al., 2017; Murugesan et al., 2024; Yoon et al., 2024). Such miscalibration can result in overconfident predictions, raising concerns about the reliability and trustworthiness of VLMs, particularly for real-world safety-critical applications that require reliable uncertainty estimates, including medical diagnostics (Ji et al., 2021; Wang et al., 2022; Zhang et al., 2023; Chen et al., 2023; Liu et al., 2023) and autonomous driving (Dorbala et al., 2022; Gadre et al., 2022; Khandelwal et al., 2022; Bucker et al., 2023; Cui et al., 2024; You et al., 2024; Zhou et al., 2024). To date, calibration in test-time prompt tuning of VLMs is less explored, with limited efforts to address it.

To do better prompt calibration, prior works, such as C-TPT (Yoon et al., 2024) and O-TPT (Sharifdeen et al., 2025) have explored methods to encourage dispersion between pairwise textual features, which can be categorized into the following two types: The first type, known as Average Textual Feature Dispersion (ATFD), spreads textual features away from their centroid. However, this can still result in textual features lying closely together (Fig. 2) and cause poor calibration performance (Fig. 1). The second type enforces orthogonality constraints to encourage angular separation, which exploits an auxiliary orthogonal regularization term in the loss function to encourage pairwise textual features as orthogonal as possible. However, we observe that it tends to group textual features closer, particularly when the number of classes $N$ is greater than the embedding dimension $|D|$ (e.g., $(N > |D|)$, where CLIP's ViT-B/16 512-d (Radford et al., 2021; Liang et al., 2022) vs. 1000 classes in ImageNet-1k, V2, K) does not guarantee uniformity of angular separation (Fig. 2, 4a). When the number of classes is less than the embedding dimension ($(N < |D|)$, e.g., classes: 10 in EuroSAT, 37 in OxfordPets, and 47 in DTD), it fails to fully utilize the hyperspherical space of feature points effectively across the hypersphere (Fig. 2, 4b). This eventually leads to poor calibration (Fig. 1). Although these methods increase feature dispersion to some extent, they often overlook the importance of angular diversity. Without sufficient angular separation, prompts may become highly correlated, which limits the model's ability to generate well-calibrated predictions. Prior work (Wang & Isola, 2020) has shown that uniformity (uniformly distributed feature points on the unit hypersphere) preserves maximal information, closely associated with strong zero-shot CLIP performance.

The uniformity problem is well-studied in Tammes problem (Tammes, 1930) (best-packing), that is to find the optimal arrangement of a given number of feature points on the surface of a unit hypersphere such that the minimum distance between any two points is maximized. Inspired by this insight, we propose **A-TPT**, a numerical optimization approach that introduces a simple yet effective angular diversity into the test-time prompt tuning framework. Our method maximizes the minimum pairwise angular distance between normalized textual features on the unit hypersphere to promote uniform and diverse prompt distribution, fully utilizing the hyperspherical space (Fig. 2). By penalizing closely aligned prompt directions, the proposed A-TPT promotes the greatest possible angular distance between them (Fig. 3b), thus achieving better prompt calibration performance (Fig. 1). Notably, by maximizing the minimum angular distance between prompt vectors, A-TPT naturally solves the number of classes exceeding the embedding dimension problem, e.g., 1000 classes in a 512-d space, making all prompt vectors pairwise orthogonal impossible. In such cases, hypersphere can still achieve a good class separation in A-TPT — maximizing angular distance leads to better performance (Fig. 4a).

Our major contributions are summarized as follows:

- We introduce a numerical optimization method, called A-TPT, for better calibration of test-time prompt tuning for VLMs. This resolves the suboptimal performance of existing leading calibration techniques for test-time prompt tuning.

- We introduce novel angular diversity that effectively promotes the diversity among textual features, thereby improving the calibration capabilities of VLMs when $N > |D|$ and $N < |D|$. This is accomplished by maximizing the minimum pairwise angular distance between normalized textual features.

- We conduct extensive experiments to validate the generalizability of our approach on different datasets, including medical datasets, across various baselines. The results show that A-TPT surpasses state-of-the-art methods in calibration performance. We also provide thorough analyses, including theoretical aspects. Moreover, our approach provides superior calibration compared to the zero-shot CLIP model, which reveals improved calibration.

## 2 RELATED WORKS

**Prompt tuning for large VLMs.** Large VLMs such as CLIP (Radford et al., 2021) and ALIGN (Jia et al., 2021) are pre-trained on extensive image-text pair datasets to learn a shared multimodal latent space (Menon & Vondrick, 2022), thereby enabling strong zero-shot performance through prompt-based inference. In large vision-language models (VLMs), predictions are guided by hand-crafted textual prompts that require domain-specific heuristics. While effective, manually designed prompts may be suboptimal across various newer domains. To address this, prompt tuning techniques treat prompts as trainable vectors and optimize them via gradient descent. Notably, CoOp (Zhou et al., 2022b) introduced a supervised prompt tuning framework for CLIP (Radford et al., 2021), which improves the classification accuracy by leveraging labeled training samples. However, follow-up work CoCoOp (Zhou et al., 2022a) showed that CoOp (Zhou et al., 2022b) struggles to generalize to out-of-distribution (OOD) data and proposed input image-conditioned prompts to enhance the model's ability to adapt to new, novel domains. Despite these advances, such methods rely on annotated training data, which limits their utility when working with pre-trained models in zero-shot settings. To address this gap, Test-time Prompt Tuning (TPT) (Shu et al., 2022) has been introduced to enable on-the-fly adaptive prompt adaptation using just one unlabelled test image sample during inference. TPT optimizes prompts by minimizing prediction entropy and boosts model accuracy in zero-shot scenarios. However, recent works (Yoon et al., 2024; Sharifdeen et al., 2025) have revealed that it leads to poorly calibrated, overconfident predictions.

**Calibration of deep neural networks.** Model calibration evaluates how well the model's predicted confidence aligns with the actual likelihood of correctness (Guo et al., 2017). Well-calibrated predictions are crucial in high-stakes applications such as healthcare, autonomous systems, and safety-critical systems (Ghahramani, 2015), where reliable uncertainty estimation is crucial for trustworthy decision-making. Calibration techniques for deep neural networks can be categorized into two categories: post-hoc and train-time methods. Post-hoc calibration strategies, such as temperature scaling (Guo et al., 2017), platt scaling (Platt et al., 1999), and conformal prediction (Vovk et al., 2005; Lei et al., 2018) calibrate a model's prediction confidence after training using a held-out validation set. However, these methods rely on access to labeled datasets collected from a distribution similar to the target data (Liu et al., 2022), often impractical in zero-shot and out-of-distribution (OOD) contexts. In contrast, train-time calibration methods integrate a hybrid calibration objective into the training of deep neural networks, with an auxiliary calibration loss as a regularizer in conjunction with the primary training loss. These include techniques (Kumar et al., 2018; Munir et al., 2022; 2023; Yoon et al., 2023) (i.e., for object classification and detection) that incorporate differentiable auxiliary regularization loss functions during training to reduce calibration error for reliable predictions. However, these train-time calibration methods are supervised and require labeled training data, limiting their applicability in the test-time prompt tuning of VLMs without supervision.

**Calibration of large VLMs.** Large VLMs such as CLIP demonstrate strong zero-shot performance by leveraging large-scale pretraining on image-text pairs. Despite the efficacy of VLMs in generalizing to new tasks, they often suffer from poor calibration. Recent works have shown that test-time prompt tuning (TPT) (Shu et al., 2022) can boost task-specific accuracy in zero-shot settings; however, it could increase the model's overconfidence by expanding the logit range during inference. To address

this, (Murugesan et al., 2024) have proposed logit normalization strategies. For example, zero-shot logit normalization adjusts the model's prediction confidence by refining the logits with (original) zero-shot baselines, while sample-adaptive logit scaling dynamically calibrates the normalized logits per instance to reduce overconfidence during inference. In parallel, C-TPT (Yoon et al., 2024) explored the relationship between textual feature dispersion and model calibration and proposed Average Text Feature Dispersion (ATFD) loss to maximize inter-class dispersion. It has been observed that well-calibrated prompts tend to generate class-specific text embeddings that are more widely separated in embedding space. To this end, the Average Text Feature Dispersion (ATFD) loss was introduced to maximize inter-class dispersion. Although ATFD effectively reduces calibration error without compromising accuracy, it may struggle in challenging cases, where it is limited in establishing enough dispersion and fails to sufficiently utilize the embedding space. To address these shortcomings, orthogonality-based regularization has been proposed (Sharifdeen et al., 2025), which enforces orthogonality constraints to encourage angular separation between textual features to promote greater dispersion. While this improves feature dispersion to some extent, it tends to group prompts closer when the number of classes is greater than the embedding dimension and does not guarantee uniform angular separation across all prompt vectors, resulting in poor calibration. Encouraged by these insights, we introduce a novel angular diversity technique that explicitly maximizes the minimum pairwise angular distance between normalized prompt vectors. This method promotes a more uniform and diverse distribution of prompts on the hypersphere, and hence better utilizes the embedding space, thereby consistently achieving better calibration performance during test-time adaptation without relying on labeled data.

## 3 PROPOSED METHOD

**Zero-shot classification with large VLMs (CLIP).** CLIP (Radford et al., 2021) consists of two encoders: an image encoder $(f_i)$ and a text encoder $(f_t)$, which map visual and textual inputs into the corresponding feature space vectors. The model is pre-trained with contrastive learning that maximizes the cosine similarity between corresponding image-text feature vectors, thereby aligning the visual and textual modalities within a shared multimodal latent space. In the zero-shot setting with CLIP, class-related textual prompts are constructed with hand-crafted templates — e.g., "`a photo of a [CLS]`" — where "`[CLS]`" corresponds to the name $c_k$ of each possible class $C = \{c_k\}_{k=1}^{N}$ from a predefined set of classes to classify images. Next, each textual prompt $\mathrm{p}_k = \mathrm{prompt}(c_k)$ corresponding to a specific class $c_k$ is fed into the text encoder $(f_t)$ to generate the textual feature vector: $t_k = f_t(\mathrm{p}_k)$. Simultaneously, a given test image $x$ is fed into the image encoder $(f_i)$ to generate the image feature vector $v = f_i(x)$. To classify the image, cosine similarities $s_k = \mathrm{sim}(v, t_k)$ are computed between the image feature vector $v$ and each class-specific text feature vector $t_k$. These similarity scores are then converted into probabilities of predicting class $c_k$ for the test image $x$ using a Softmax function controlled by a temperature parameter $\tau$, which is fixed at 0.01 during inference. Then, the predicted class becomes the class with the highest probability, $\hat{c} = \arg\max_{c_k} p(c_k \mid x)$ with its associated predicted confidence is $\hat{p} = \max_{c_k} p(c_k \mid x)$. This zero-shot framework enables efficient classification across a wide range of categories without additional fine-tuning, relying solely on the learned multimodal alignment between images and textual prompts. In contrast to hand-crafted prompts (i.e., hard prompts), prompt tuning has been explored in CLIP (Chen et al., 2022; Radford et al., 2021; Yao et al., 2023; Zhou et al., 2022b;a) to optimize trainable prompt embeddings using 16 samples per class from the ImageNet dataset, which allows the learned prompts to generalize across cross-datasets. Recently, TPT (Shu et al., 2022) has enabled prompt tuning without labeled data during inference. Although it boosts accuracy, TPT often raises calibration errors due to overconfident predictions (Guo et al., 2017).

**Expected calibration error and evaluation metric.** Lower expected calibration error (ECE) (Naeini et al., 2015) indicates perfect calibration, where the model ensures that predicted probabilities correspond accurately to the likelihood of true accuracy. We define this as: $\mathbb{P}(\hat{c} = C \mid \hat{p} = p) = p, \forall p \in [0, 1]$, where an input image $x$ with its corresponding ground truth label $c$, predicted class label $\hat{c}$ with its predicted confidence $p = \hat{p}$. Expected Calibration Error (ECE) partitions predictions into bins based on the confidence metric and evaluates the absolute difference between the accuracy and the mean confidence within each bin. Mathematically, ECE is formulated as follows: $\mathrm{ECE} = \sum_{n=1}^{N} \frac{|B_n|}{M} |\mathrm{acc}(B_n) - \mathrm{conf}(B_n)|$, where $N$ denotes the total number of bins, $B_n$

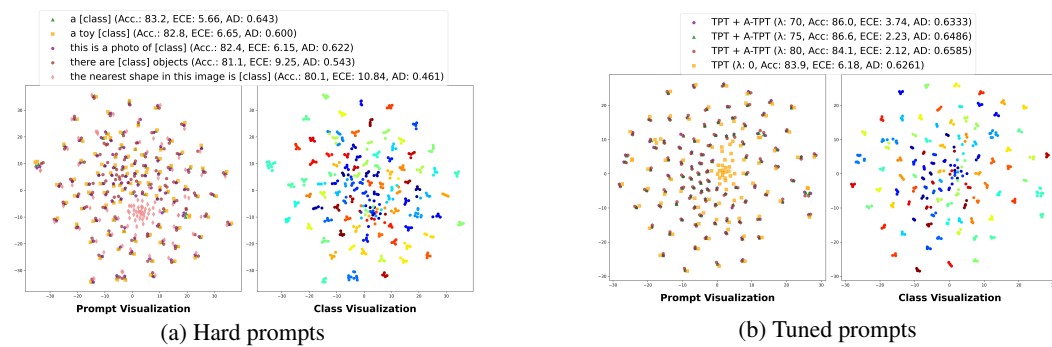

Figure 3: **t-SNE visualization of class-wise embedded textual features** with CLIP RN50 model on the fine-grained classification dataset (Fei-Fei et al., 2004) for (a) hard prompts and (b) tuned prompts. In both subfigures, each unique color represents a distinct prompt in the prompt visualization (left) and a distinct class in the class visualization (right). The legends belong to the prompt visualization (left) for both subfigures.

defines the image set with predicted confidence falling into the $n$-th bin, $|B_n|$ is the number of images within this bin, and $M$ is the total number of predictions, Furthermore, acc $(B_n)$, and conf $(B_n)$ represents the accuracy of the predictions and average prediction confidence of all associated with bin $n$, respectively.

**Why is angular diversity better for prompt calibration?** While different prompts may yield text dispersion that results in comparable classification accuracies, their calibration performance can vary significantly. To further understand the relationship between angular diversity — maximizing the angular distance (AD) and expected calibration error (ECE) — we conduct experiments with 80 different hard prompt styles (Radford et al., 2021). We followed the C-TPT (Yoon et al., 2024) to evaluate the impact of angular distance (AD) and calibration error within the same accuracy group. Specifically, we focus on the 3 well-calibrated ('a', 'a toy', 'this is a photo of') and 2 poor-calibrated ('there are [CLS] objects', 'the nearest shape in this image is') prompt styles that provide lower and higher ECEs. For illustration, consider the following examples from the Caltech101 dataset, using CLIP RN50 are categorized into well-calibrated and poor-calibrated prompts:

**Hard prompts** (See Fig. 3a legend)

○ a [CLS] - Acc: 83.2, ECE: 5.66, AD: 0.643

○ a toy [CLS] - Acc: 82.8, ECE: 6.65, AD: 0.600

○ this is a photo of [CLS] - Acc: 82.4, ECE: 6.15, AD: 0.622

○ there are [CLS] objects - Acc: 81.1, ECE: 9.25, AD: 0.543

○ the nearest shape in this image is [CLS] - Acc: 80.1, ECE: 10.84, AD: 0.461

**Tuned prompts** (See Fig. 3b legend)

○ a toy [CLS]: TPT - Acc: 83.9, ECE: 6.18, AD: 0.6216

○ this is a photo of [CLS]: TPT + A-TPT - Acc: 86.0, ECE: 3.74, AD: 0.6333

○ a [CLS]: TPT + A-TPT - Acc: 86.6, ECE: 2.23, AD: 0.6486

○ there are [CLS] objects: TPT + A-TPT - Acc: 84.1, ECE: 2.12, AD: 0.6585

While well-calibrated prompts typically yield higher accuracy, the calibration error varies significantly within the same accuracy group. Similarly, poor-calibrated prompts tend to yield lower accuracy, and the calibration error varies significantly within the group. That is why in our analysis in Fig. 3, we collected the prompts that yielded similar accuracy and tried to determine what caused the difference in the calibration error within the same accuracy group.

We visualize t-SNE of class-wise embedded textual features in Fig. 3. In the Prompt Visualization (*left*), each point represents a text feature from different prompts for all possible classes. In the Class Visualization (*right*), the class embeddings represent text features corresponding to the actual class labels. These visualizations illustrate the distribution of different hard prompt vectors in the feature space, providing insights into how the angular diversity affects model calibration. In the case of Fig. 3a (*left*), we observed a distinct pattern; poorly calibrated prompts — low angular diversity — are clustered closely together regardless of their corresponding class labels. This low angular diversity results in poor calibration, as the text features become highly correlated. In contrast, well-calibrated prompts — high angular diversity — are well dispersed across the feature space, as shown in Fig. 3a (*left*). Notably, the features for well-calibrated prompts tend to group cohesively associated with their class labels (Fig. 3a (*right*)). These well-dispersed features allow the model to achieve better calibration, as the features align closely with their respective class label locations in the feature space.

This pattern suggests that angular diversity disperses class-specific prompts and clusters text features near their corresponding class labels, reducing calibration errors. In Fig. 3b, when applying Test-time Prompt Tuning (TPT), the text features tend to cluster together, similar to the poorly calibrated hard prompts. However, as we introduce angular diversity (A-TPT) and progressively increase the regularization strength during the optimization process (via $\lambda$), the angular distance between the features increases. As shown in Fig. 3b (*left*), A-TPT results in greater angular diversity than TPT alone. Notably, the features align more closely with their respective class labels (Fig. 3b (*right*)), similar to the well-calibrated hard prompt scenario. As shown in Figure 4, the examples provided show a negative correlation between ECE and angular diversity — maximizing the angular distance (AD) within each group, which aligns with the empirical findings of our paper. These findings suggest the importance of angular diversity in improving VLM calibration for test-time prompt tuning. By increasing the angular distance between learned prompt vectors, A-TPT promotes well-calibrated predictions, crucial for real-world applications that require reliable uncertainty estimation, such as medical diagnostics and autonomous systems.

**Motivation for angular diversity.** Prior works have shown that well-calibrated textual features tend to be more dispersed with L2 distance (Yoon et al., 2024), which is spread farther apart or benefits from greater angular separation with enforced orthogonality constraints (Sharifdeen et al., 2025) in the embedding space. Prior work O-TPT (Sharifdeen et al., 2025) found that lower cosine similarity between textual features corresponds to better model calibration. This suggests that promoting greater angular separation among class-wise text features can improve the calibration of VLMs. However, these analyses overlook a crucial aspect: angular diversity among textual features. We empirically observe that orthogonalization tends to group textual features closer, particularly when the number of classes is greater than the embedding dimension (Fig. 2, 4a), and does not guarantee uniformity of angular separation across all prompt vectors. We hypothesize that promoting angular diversity is more important for better calibration of VLMs than enforcing orthogonality among textual features. Unlike existing techniques that disperse prompts via L2 dispersion (Yoon et al., 2024) or orthogonality constraints (Sharifdeen et al., 2025), our approach explicitly focuses on maximizing the minimum pairwise angular distance among normalized textual features. This numerical method constructs each prompt-induced feature point uniformly distributed on a unit hypersphere. Maximizing the minimum pairwise angular distance between the features ensures each feature vector points in a diverse direction, thereby sharpening class boundaries and improving zero-shot calibration performance in VLMs (Wang & Isola, 2020) during inference.

**Comparison of dispersion, orthogonality, and angular diversity.** The ATFD (Yoon et al., 2024) objective disperses textual features by maximizing the L2 distance from their centroid, without enforcing pairwise angular separation. In contrast, orthogonality constraints (Sharifdeen et al., 2025) that enforce angular separation by pushing features to be orthogonal to minimize their cosine similarity, tend to group textual features closer together when the number of classes is greater than the embedding dimension $N > |D|$. When the number of classes is lesser $N < |D|$, it fails to fully utilize the hyperspherical space effectively. As a result, neither of these methods guarantees the uniformity of angular separation of features across the hypersphere (Liang et al., 2022). In zero-shot CLIP settings, normalized textual features are constrained to the surface of a unit hypersphere (Wang & Isola, 2020). While ATFD may shift the centroid of the text features toward the center of the hypersphere by adjusting the features, similarly, orthogonality constraints may encourage angular separation by pairwise orthogonalizing the features on the surface. However, these methods may not effectively distribute features uniformly across the hypersphere's surface. To further validate our findings, we experimented for both where $N < |D|$ (Helber et al., 2018), and $N > |D|$ (Recht et al., 2019) cases. For each data sample, we extract test-time prompt-tuned text features generated by O-TPT and A-TPT (ours), compute pairwise cosine similarities, and plot their mean, as shown in Fig. 4. The results show that O-TPT, which lacks calibration-specific constraints, tends to group textual features closer when $N > |D|$ (O-TPT fails), exhibits lower, but high fluctuations in cosine similarities, reflecting its inconsistent calibration performance. In contrast, our method's angular diversity consistently produces text features with slightly higher but more consistent cosine similarities, indicating stable, uniform angular separation. Similarly, O-TPT underutilizes hyperspherical space, showing higher, slightly fluctuating cosine similarities when $N < |D|$, while A-TPT shows lower, more consistent cosine similarities, achieving the greatest possible angular separation. That's why hypersphere offers optimal textual feature separation in A-TPT. (suppl. carries more details) In Tab. 1 we present the accuracy and ECE results for CLIP ViT-B/16 backbone, dividing data into two groups based on the number of classes relative to the embedding dimension of TPT text features. Group 1 includes

| Method | Metric | Group 1 ($N > |D|$) | Group 2 ($N < |D|$) | Overall |
|---|---|---|---|---|
| Baseline | Acc. | 57.87 | 62.99 | 60.43 |
| | ECE | 3.36 | 5.44 | 4.40 |
| TPT | Acc. | 59.83 | 64.54 | 62.19 |
| | ECE | 12.60 | 9.89 | 11.25 |
| C-TPT | Acc. | 59.70 | 64.44 | 62.07 |
| | ECE | 5.58 | 5.25 | 5.42 |
| O-TPT | Acc. | 58.70 | 63.63 | 61.17 |
| | ECE | 4.27 | 4.44 | 4.36 |
| **A-TPT (Ours)** | **Acc.** | 58.23 | 64.30 | 61.27 |
| | **ECE** | 2.92 | 3.60 | 3.26 |

Table 1: Comparison of Accuracy and ECE across methods with CLIP ViT-B/16 backbone and categories based on the number of classes and TPT text features embedding dimension ($|D|$).

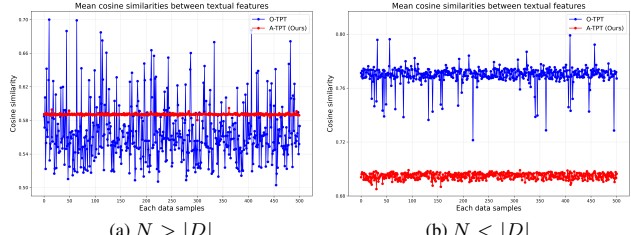

(a) $N > |D|$        (b) $N < |D|$

Figure 4: Comparison of mean cosine similarity changes for both categories with CLIP ViT-B/16 backbone. Where, O-TPT fails, but our A-TPT offers consistent cosine similarity values and achieves the greatest minimum pairwise angular distance among text features for all the data points. (suppl. carries more details.)

cases with $N > |D|$, while Group 2 includes $N < |D|$. We then calculate the ECE and accuracy separately for each group, allowing a more fine-grained analysis of each method's performance. As hypothesized, cases with $N > |D|$ tend to show elevated ECE, indicating poor calibration and suggesting these are more challenging points. In these challenging cases (Group 1), our method significantly outperforms C-TPT as well as O-TPT in terms of calibration performance, resulting in an overall lower ECE. These results highlight the efficacy of our approach in handling both groups. To achieve well-calibrated predictions, we argue that simple feature dispersion or orthogonal separation is insufficient, instead promoting angular diversity — by maximizing the minimum pairwise angular distance — an effective approach to uniformly distributing features across the hypersphere's surface and improving calibration in VLMs.

**Angular diversity.** Motivated by these insights, we introduce angular diversity to better the test-time prompt calibration of VLMs by promoting angular diversity within the textual feature matrix. Let each class $c_k$ be associated with a textual feature vector $t_k \in \mathbb{R}^{|D|}$, where $|D|$ is the embedding dimension. Define the text feature matrix $\mathbf{E}$ that contains textual feature vectors for all classes, such that $\mathbf{E} \in \mathbb{R}^{N \times |D|}$, where $N$ denotes the total number of classes. Each element $\mathbf{E}_{ij}$ corresponds to the embedding of the $i$-th class in the $j$-th dimension. This matrix $\mathbf{E}$ captures the spatial distribution of class-specific features across the shared latent space. We normalize $\mathbf{E}$ to $\hat{\mathbf{E}}$. To promote uniformity on a unit hypersphere (Fig. 2), we compute the matrix product $\hat{\mathbf{E}}\hat{\mathbf{E}}^T$, which contains the pairwise cosine similarities between the text features. Inspired by insights from the ArcFace (Deng et al., 2019), we propose an angular variant of the cosine loss as the objective function to maximize the minimum pairwise angular distance between the normalized prompt vectors.

$$\text{AD} = \frac{1}{N} \sum_{i=1}^{N} \min_{j \in \{1,...,N\} \setminus \{i\}} \boldsymbol{\theta}_{ij}, \quad \boldsymbol{\theta} = \arccos(\hat{\mathbf{E}}\hat{\mathbf{E}}^T), \quad \text{s.t.} \, \# \, \forall_i \, \hat{\mathbf{E}}_i = \frac{\mathbf{e}_i^T}{|\mathbf{e}_i|}, \qquad (1)$$

Here, $\boldsymbol{\theta} \in \mathbb{R}^{N \times N}$ is the matrix of pairwise angular distances. The angular diversity term, denoted as AD, maximizes the minimum pairwise angular distance between normalized prompt vectors while ensuring uniformity of text features across the feature space. Thus, we integrate this regularization term into the overall objective function for the test-time prompt tuning process to better the calibration performance is formulated as:

$$\mathbf{p}^* = \arg\min_{\mathbf{p}} \left( \mathcal{L}_{\text{TPT}} + \lambda \cdot \mathcal{L}_{\text{A-TPT}} \right), \quad \text{where } \mathcal{L}_{\text{A-TPT}} = -\text{AD}, \qquad (2)$$

$\mathcal{L}_{\text{TPT}}$ is the TPT negative maximum class log probability (entropy minimization) loss function, $\mathbf{p}$ denotes the learnable prompt parameters, and $\lambda$ is a hyperparameter that promotes uniformity of distributed features across the hypersphere and ensures effective utilization of the full feature space while controlling the strength of the regularization term. By explicitly promoting the angular diversity term, we systematically achieve maximum angular distance between pairwise textual features to better prompt calibration in test-time prompt tuning.

**Numerical stability analyses.** When computing $\arccos$ which could suffer from gradient explosion if its input leaks outside the boundaries of $[-1, 1]$ and gets too close to the ends (as its gradient is $\frac{\partial}{\partial x}(\arccos(x)) = -1/\sqrt{(1-x^2)}$, In Eq. 1, where $\boldsymbol{\theta}$ is the arccos of pairwise cosine similarities (dot products of normalized text features $\hat{\mathbf{E}}\hat{\mathbf{E}}^T$), we clamp within (-1, 1), specifically from -0.99999 to 0.99999, to prevent NaNs in the $\arccos$ calculation, and to prevent the infinite gradient at $\pm 1$ during backpropagation.

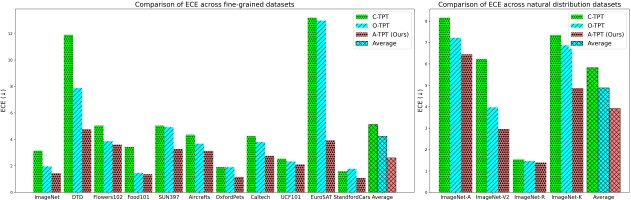 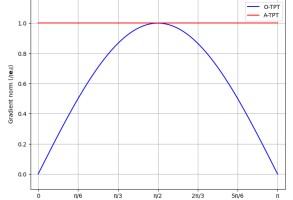

Figure 6: Comparison of expected calibration error (ECE) between C-TPT, O-TPT, and A-TPT (Ours). Results are based on CLIP ViT-B/16 backbone. Lower ECE provides better prompt calibration.

Figure 7: Comparison of the gradient norm changed with pairwise angular distance. Unlike O-TPT's gradient, which vanishes as $\theta \to 0$, A-TPT's gradient is stable and consistent with $\theta$.

**Gradient analyses comparison to O-TPT.** O-TPT's orthogonality loss yields gradients that shrink to zero as the pairwise angular distance $\theta \to 0$, making optimization hard when features are already close. In contrast, A-TPT optimizes the angular distance directly; its gradient norm is angle-independent, so it stays stable even at small $\theta$. That's why directly optimizing angular distance rather than using cosine similarity at test time improves VLM calibration and avoids the stuck near-colinear regime that hurts calibration. (See Fig. 7 for gradient-norm, and Appendix A.5 for derivations (eqs. (3)–(4)).

**Computational complexity.** A-TPT's asymptotic complexity same as O-TPT, with negligible run-time/memory overhead over C-TPT, while substantially reducing ECE. (See Tab. 8 and Appendix A.6)

## 4 EXPERIMENTS

We evaluate on different datasets, across various baselines, with CLIP ViT-B/16 (512-d) and RN50 (1024-d); suppl. carries datasets and implementation details in Appendix A.7 and A.9.

**Calibration performance on fine-grained classification tasks.** We evaluate the proposed A-TPT method, and we observe better calibration performance across multiple fine-grained classification tasks with both CLIP ViT-B/16 and CLIP RN50 backbones (Tab. 2. A-TPT consistently reduces ECE compared to O-TPT (Sharifdeen et al., 2025) and C-TPT (Shu et al., 2022): For CLIP ViT-B/16, average ECE drops from 5.13 (C-TPT) and 4.23 (O-TPT) to **2.92**. For CLIP RN50, ECE reduces from 6.19 (C-TPT) and 5.45 (O-TPT) to **2.79** These results highlight the efficacy of A-TPT in both $N < |D|$ and $N > |D|$ cases.

**Calibration performance under natural distribution shifts.** Tab. 3 shows the calibration results under natural distribution shifts. All hyperparameters and experimental configurations match with the implementation section, except for the regularization weight $\lambda$, which we set to 10.0. Similar to Tab. 2, A-TPT shows better calibration performance across ImageNet variants by reducing the ECE for both CLIP ViT-B/16 and CLIP RN50. For CLIP ViT-B/16, A-TPT lowers the average ECE to **3.92**, drops from 4.88 (O-TPT) and 5.82 (C-TPT). For CLIP RN50, A-TPT achieves an average ECE of **7.82** drops from 9.69 (O-TPT) and 12.1 (C-TPT). Importantly, A-TPT also surpasses the zero-shot baseline in calibration performance, showing lower ECE on both backbones, without compromising the high accuracy benefits of TPT for $N > |D|$ and $N < |D|$ cases, which is a feat unmatched by any other approach.

**Medical prompt tuning with A-TPT.** We evaluate the generalizability of A-TPT on medical datasets with medical baselines under the $N < |D|$ regime. Tab. 4 presents the performance of FPT (Huang et al., 2024) and FPT combined with O-TPT and A-TPT on ISIC 2018 dataset, where A-TPT leads to a notable reduction in Expected Calibration Error (ECE) while preserving high classification accuracy. Tab. 5 evaluates PLIP with Prompt Smooth (PS) (Hussein et al., 2024), on KatherColon, where the combination of A-TPT further better calibration performance. Tab. 6 reports results using MedCLIP with BAPLe (Hanif et al., 2024), showing that the integration of A-TPT substantially improves calibration metrics over the baseline.

**Comparison with previous calibration methods.** As illustrated in Fig. 6, average ECE across different datasets, including fine-grained classification and natural distribution shifts, where calibration techniques are applied to TPT (Shu et al., 2022). We compare C-TPT (Yoon et al., 2024), O-TPT (Sharifdeen et al., 2025), and A-TPT (Ours). A-TPT consistently shows better calibration, demonstrating its efficacy in improving model reliability across datasets.

**Reliability plots.** To address under-confidence and over-confidence, while Tables 2 and 3 illustrate that A-TPT achieves superior calibration performance across multiple fine-grained datasets on both

| Method | Metric | ImageNet | DTD | Flowers102 | Food101 | SUN397 | Aircrafts | OxfordPets | Caltech101 | UCF101 | EuroSAT | Standford Cars | Average |
|---|---|---|---|---|---|---|---|---|---|---|---|---|---|
| **Pre-trained Backbone: CLIP ViT-B/16** | *Embedding dimension: 512-d* | | | | | | | | | | | | |
| Baseline | Acc. | 66.70 | 44.30 | 67.30 | 83.60 | 62.50 | 23.90 | 88.00 | 92.90 | 65.00 | 41.30 | 65.30 | 63.70 |
| | ECE | 2.12 | 8.50 | 3.00 | 2.39 | 2.53 | 5.11 | 4.37 | 5.50 | 3.59 | 13.89 | 4.25 | 4.43 |
| TPT | Acc. | 69.00 | 46.70 | 69.00 | 84.70 | 64.50 | 23.40 | 87.10 | 93.80 | 67.30 | 42.40 | 66.30 | 65.00 |
| | ECE | 10.60 | 21.20 | 13.50 | 3.98 | 11.30 | 16.80 | 5.77 | 4.51 | 2.54 | 13.20 | 5.16 | 11.60 |
| C-TPT | Acc. | 68.50 | 46.00 | 69.80 | 83.70 | 64.80 | 24.85 | 88.20 | 93.63 | 65.70 | 43.20 | 65.80 | 64.57 |
| | ECE | 3.15 | 11.90 | 5.04 | 3.43 | 5.04 | 4.36 | 1.90 | 4.24 | 2.54 | 13.20 | 1.59 | 5.13 |
| O-TPT | Acc. | 67.33 | 45.68 | 70.07 | 84.13 | 64.23 | 23.64 | 87.95 | 93.95 | 64.16 | 42.84 | 64.53 | 64.41 |
| | ECE | 1.96 | 7.88 | 3.87 | 1.46 | 4.93 | 3.68 | 1.90 | 2.07 | 2.34 | 12.98 | 1.78 | 4.23 |
| **A-TPT (Ours)** | **Acc.** | 67.70 | 45.51 | 69.22 | 83.64 | 66.04 | 23.76 | 88.33 | 93.87 | 66.16 | 44.06 | 65.78 | 64.92 |
| | **ECE** | 1.45 | 4.76 | 3.61 | 1.37 | 3.28 | 3.14 | 1.17 | 2.76 | 2.12 | 3.92 | 1.09 | **2.61** |
| **Pre-trained Backbone: CLIP RN50** | *Embedding dimension: 1024-d* | | | | | | | | | | | | |
| Baseline | Acc. | 58.10 | 40.00 | 61.00 | 74.00 | 58.60 | 15.60 | 83.80 | 85.80 | 58.40 | 23.70 | 55.70 | 55.90 |
| | ECE | 2.09 | 9.91 | 3.19 | 3.11 | 3.54 | 6.45 | 5.91 | 3.05 | 3.33 | 15.40 | 4.70 | 5.61 |
| TPT | Acc. | 60.70 | 41.50 | 62.50 | 74.90 | 61.10 | 17.00 | 84.50 | 87.00 | 59.50 | 28.30 | 58.00 | 57.70 |
| | ECE | 11.40 | 25.70 | 13.40 | 5.25 | 9.24 | 16.10 | 3.65 | 5.04 | 12.40 | 22.50 | 3.76 | 11.70 |
| C-TPT | Acc. | 60.20 | 42.20 | 65.20 | 74.70 | 61.00 | 17.00 | 84.10 | 86.90 | 59.70 | 27.80 | 56.50 | 57.75 |
| | ECE | 3.01 | 19.80 | 4.14 | 1.86 | 2.93 | 10.70 | 2.77 | 2.07 | 3.83 | 15.10 | 1.94 | 6.19 |
| O-TPT | Acc. | 58.97 | 41.90 | 65.61 | 74.22 | 60.85 | 16.77 | 83.40 | 86.86 | 58.84 | 28.35 | 56.44 | 57.47 |
| | ECE | 3.10 | 16.53 | 2.50 | 1.20 | 3.20 | 8.18 | 3.50 | 2.75 | 2.60 | 14.71 | 1.69 | 5.45 |
| **A-TPT (Ours)** | **Acc.** | 58.44 | 40.90 | 64.89 | 74.10 | 60.46 | 14.58 | 83.48 | 86.57 | 60.24 | 32.14 | 57.08 | 57.53 |
| | **ECE** | 2.49 | 6.41 | 2.39 | 1.11 | 2.90 | 6.14 | 2.47 | 1.98 | 2.34 | 2.51 | 1.38 | **2.92** |

Table 2: Comparison of methods across fine-grained datasets for Accuracy (Acc.) and Expected Calibration Error (ECE) with CLIP ViT-B/16 and CLIP RN50 pre-trained backbone for both $N > |D|$ and $N < |D|$ cases. The overall top best-performing result is in bold.

| Method | Metric | ImageNet-A | ImageNet-V2 | ImageNet-R | ImageNet-S | Average |
|---|---|---|---|---|---|---|
| **Pre-trained Backbone: CLIP ViT-B/16** | *Embedding dimension: 512-d* | | | | | |
| Baseline | Acc. | 47.80 | 60.80 | 74.00 | 46.10 | 57.20 |
| | ECE | 8.61 | 3.01 | 3.58 | 4.95 | 5.04 |
| TPT | Acc. | 52.60 | 63.00 | 76.70 | 47.50 | 59.90 |
| | ECE | 16.40 | 11.10 | 4.36 | 16.10 | 12.00 |
| C-TPT | Acc. | 51.60 | 62.70 | 76.00 | 47.90 | 59.60 |
| | ECE | 8.16 | 6.23 | 1.54 | 7.35 | 5.82 |
| O-TPT | Acc. | 49.87 | 61.65 | 72.55 | 47.12 | 57.80 |
| | ECE | 7.22 | 3.97 | 1.46 | 6.87 | 4.88 |
| **A-TPT (Ours)** | **Acc.** | 50.39 | 60.90 | 74.87 | 46.09 | 58.06 |
| | **ECE** | 6.45 | 2.96 | 1.39 | 4.87 | **3.92** |
| **Pre-trained Backbone: CLIP RN50** | *Embedding dimension: 1024-d* | | | | | |
| Baseline | Acc. | 21.70 | 51.40 | 56.00 | 33.30 | 40.60 |
| | ECE | 21.30 | 3.33 | 2.07 | 3.15 | 7.46 |
| TPT | Acc. | 25.20 | 54.60 | 58.90 | 35.10 | 43.50 |
| | ECE | 31.00 | 13.10 | 9.18 | 13.70 | 16.70 |
| C-TPT | Acc. | 23.40 | 54.70 | 58.00 | 35.10 | 42.80 |
| | ECE | 25.40 | 8.58 | 4.57 | 9.70 | 12.10 |
| O-TPT | Acc. | 23.07 | 53.11 | 54.47 | 33.98 | 41.16 |
| | ECE | 24.56 | 3.87 | 4.47 | 5.85 | 9.69 |
| **A-TPT (Ours)** | **Acc.** | 21.66 | 51.48 | 55.78 | 33.37 | 40.57 |
| | **ECE** | 21.14 | 3.10 | 3.96 | 3.09 | **7.82** |

Table 3: Comparison of methods across natural distribution shift datasets for Accuracy (Acc.) and Expected Calibration Error (ECE) with TPT (baseline) with CLIP ViT-B/16 (top) and CLIP RN50 (bottom) pre-trained backbones for both $N > |D|$ and $N < |D|$ cases. The overall top best-performing result is in bold.

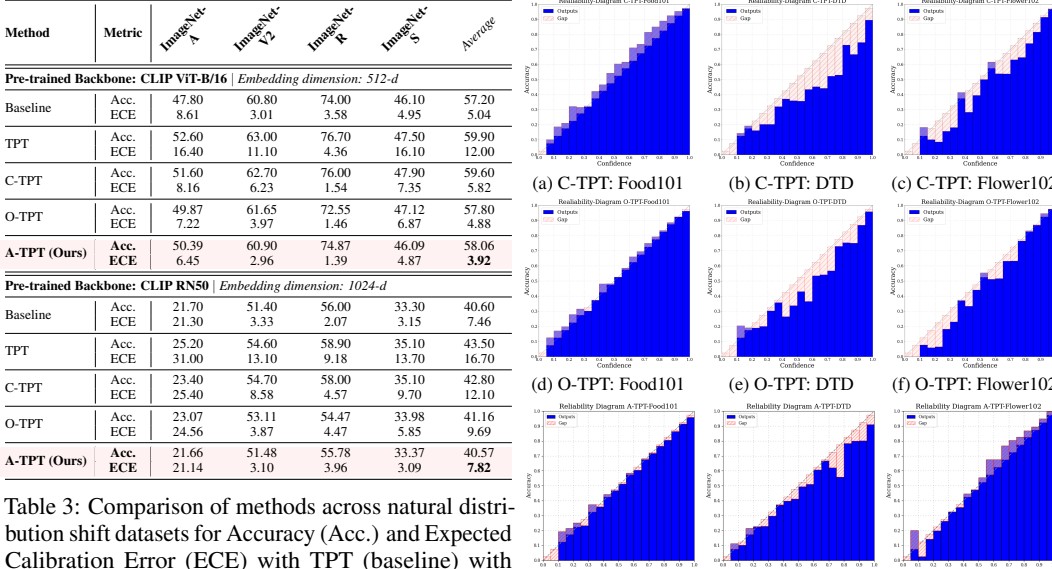

(a) C-TPT: Food101    (b) C-TPT: DTD    (c) C-TPT: Flower102

(d) O-TPT: Food101    (e) O-TPT: DTD    (f) O-TPT: Flower102

(g) A-TPT: Food101    (h) A-TPT: DTD    (i) A-TPT: Flower102

Figure 5: Reliability diagrams for CLIP ViT-B/16 backbone (suppl. carries additional reliability diagrams).

| Method | Metric | ISIC'18 ($N = 7$) |
|---|---|---|
| FPT (512-d) | Acc. | 98.43 |
| | ECE | 0.2328 |
| FPT + O-TPT | Acc. | 98.25 |
| | ECE | 0.1381 |
| **FPT + A-TPT** | **Acc.** | 98.31 |
| | **ECE** | **0.0794** |

Table 4: **FPT:** FPT + A-TPT on ISIC 2018.

| Method | Metric | KatherColon ($N = 9$) |
|---|---|---|
| PS (768-d) | Acc. | 76.6 |
| | ECE | 15.54 |
| PS + O-TPT | Acc. | 76.2 |
| | ECE | 12.73 |
| **PS + A-TPT** | **Acc.** | 76.4 |
| | **ECE** | **8.86** |

Table 5: **PLIP:** Promptsmooth (PS) + A-TPT on KatherColon (KC).

| Method | Metric | Covid BA ($N = 2$) | Covid CA ($N = 10$) |
|---|---|---|---|
| BAPLe (768-d) | Acc. | 99.90 | 82.5 |
| | ECE | 3.21 | 15.64 |
| BAPLe + O-TPT | Acc. | 99.62 | 81.36 |
| | ECE | 0.91 | 5.97 |
| **BAPLe + A-TPT** | **Acc.** | 99.78 | 82.19 |
| | **ECE** | **0.42** | **2.34** |

Table 6: **MedCLIP:** BAPLe + A-TPT on Covid dataset.

CLIP ViT-B/16 and CLIP RN-50 backbones, they do not reveal insights into whether the model tends to be over-confident or under-confident. To further analyze this, we plot reliability diagrams in Fig. 5 for the Food101, DTD, and Flowers102 datasets with the CLIP ViT-B/16 backbone. C-TPT (Yoon et al., 2024) displays under-confidence on Food101 (Fig. 5a) and over-confidence on DTD (Fig. 5b) and Flowers102 (Fig. 5c). O-TPT (Sharifdeen et al., 2025) partially mitigates these issues to some

| Method | Metric | DTD | Flowers102 | Food101 | SUN397 | Aircrafts | OxfordPets | Caltech101 | UCF101 | EuroSAT | Standford Cars | Average |
|---|---|---|---|---|---|---|---|---|---|---|---|---|
| **Pre-trained Backbone: CLIP ViT-B/16 | Baseline + CoOp | *Embedding dimension: 512-d*** | | | | | | | | | | | | |
| Baseline + CoOp | Acc. | 43.10 | 67.40 | 83.20 | 63.70 | 18.00 | 89.20 | 93.60 | 66.00 | 40.10 | 63.10 | 63.50 |
| | ECE | 7.71 | 3.92 | 1.55 | 1.72 | 9.21 | 2.92 | 3.65 | 3.47 | 15.30 | 6.86 | 5.25 |
| TPT + CoOp | Acc. | 44.50 | 68.70 | 83.80 | 65.60 | 20.00 | 89.10 | 94.00 | 67.20 | 40.60 | 65.60 | 63.91 |
| | ECE | 34.80 | 19.90 | 9.66 | 20.80 | 29.60 | 7.40 | 3.65 | 19.90 | 31.30 | 6.63 | 18.36 |
| TPT + CoOp + C-TPT | Acc. | 45.00 | 69.00 | 83.70 | 65.10 | 19.20 | 89.30 | 93.90 | 66.60 | 40.70 | 63.10 | 63.56 |
| | ECE | 21.00 | 10.20 | 4.49 | 11.80 | 21.50 | 2.12 | 1.66 | 12.00 | 13.20 | 2.45 | 10.04 |
| TPT + CoOp + O-TPT | Acc. | 45.45 | 68.57 | 83.55 | 64.01 | 18.69 | 89.07 | 93.71 | 65.64 | 40.17 | 64.12 | 63.14 |
| | ECE | 16.02 | 6.81 | 3.59 | 7.23 | 16.82 | 1.92 | 0.92 | 9.16 | 13.76 | 2.85 | 7.91 |
| **TPT + CoOp + A-TPT** | **Acc.** | 43.21 | 68.94 | 83.23 | 65.34 | 20.58 | 90.02 | 93.23 | 69.99 | 40.28 | 65.89 | 64.07 |
| | **ECE** | 6.33 | 2.91 | 3.12 | 2.63 | 5.51 | 1.06 | 1.08 | 3.78 | 7.85 | 2.04 | **3.63** |
| **Pre-trained Backbone: CLIP ViT-B/16 | Baseline + CoCoOp | *Embedding dimension: 1024-d*** | | | | | | | | | | | | |
| Baseline + CoCoOp | Acc. | 44.60 | 68.40 | 84.10 | 63.00 | 24.20 | 88.30 | 91.00 | 67.00 | 44.10 | 64.90 | 64.30 |
| | ECE | 3.82 | 3.82 | 3.25 | 4.61 | 4.06 | 4.60 | 3.28 | | 5.81 | 6.51 | 4.20 |
| TPT + CoCoOp | Acc. | 45.00 | 68.60 | 84.60 | 64.00 | 24.90 | 88.50 | 91.20 | 67.80 | 44.50 | 65.90 | 64.90 |
| | ECE | 6.91 | 4.70 | 1.94 | 3.16 | 6.13 | 2.22 | 2.74 | 3.47 | 9.03 | 5.22 | 4.35 |
| TPT + CoCoOp + C-TPT | Acc. | 44.70 | 69.30 | 84.20 | 63.60 | 24.60 | 88.80 | 91.40 | 67.10 | 44.30 | 64.90 | 64.70 |
| | ECE | 4.18 | 3.13 | 2.66 | 2.96 | 4.90 | 3.76 | 3.45 | 2.91 | 5.79 | 5.09 | 3.68 |
| **TPT + CoCoOp + A-TPT** | **Acc.** | 44.28 | 68.73 | 84.12 | 63.68 | 24.14 | 88.37 | 91.21 | 67.89 | 44.16 | 64.91 | 64.15 |
| | **ECE** | 3.52 | 3.08 | 1.91 | 2.74 | 4.79 | 2.09 | 2.67 | 2.85 | 3.49 | 4.95 | **3.22** |

Table 7: Comparison of methods when using CoOp (Zhou et al., 2022b) and CoCoOp (Zhou et al., 2022a) as a baseline across fine-grained datasets for Accuracy (Acc.) and Expected Calibration Error (ECE) with CLIP ViT-B/16 pre-trained backbone. The overall best-performing result is in bold.

extent (Figs. 5d, 5e, 5f), but noticeable calibration gaps persist, particularly on DTD (Fig. 5e). In contrast, A-TPT produces the most reliable predictions — correcting under-confidence in Food101 (Fig. 5g) and significantly reducing over-confidence in the other datasets (Figs. 5h, 5i). These results highlight that A-TPT effectively balances confidence and accuracy, making it an ideal solution for both under-confidence and over-confidence in VLM calibration.

**Supervised-trained prompt embeddings.** In addition to tuning prompt parameters during inference time, we further evaluate the calibration capabilities of A-TPT when combined with supervised-trained prompt embedding parameters for test-time prompt tuning. Specifically, utilized the officially published checkpoints of CoOp (Zhou et al., 2022b) and CoCoOp (Zhou et al., 2022a) as presented in Tab. 7. For CoCoOp, we trained on 16 images per class from half of ImageNet's total classes with 4 learnable prompt embeddings (Shu et al., 2022; Yoon et al., 2024). As reported in Tab. 7, for CoOp across 10 fine-grained datasets, A-TPT reduces the overall average ECE to **3.63**. This reflects a marked improvement relative to O-TPT + CoOp, C-TPT + CoOp, and TPT + CoOp, which report average ECEs of 7.91, 10.4, and 18.36, respectively. Similarly, when integrating A-TPT with CoCoOp results in considerable calibration gains across most datasets compared to C-TPT + CoCoOp and TPT + CoCoOp. As in Tab. 7, our approach reduces the overall ECE to **3.22**, compared to 3.68, 4.35 for C-TPT + CoCoOp, TPT + CoCoOp, respectively. These results show that integrating A-TPT with supervised-trained prompt embeddings during test-time tuning preserves classification accuracy while substantially better calibration performance.

## 5 CONCLUSION

We propose a novel technique, called angular diversity, to promote the uniformity of textual features and disperse them to calibrate test-time prompt tuning of vision-language models. We reveal that maximizing the minimum pairwise angular distance between textual features while prompt learning is associated with lower calibration error. We show that achieving uniformity between textual features is more effective than orthogonalization and dispersion through L2 distance objectives. Moreover, angular diversity is also effective for prompt learning with significant margins, due to enlarging the inter-class separability. Therefore, we propose angular diversity on textual features during test-time prompt tuning, abbreviated as A-TPT. Our approach consistently outperforms state-of-the-art methods with different backbones and baselines.

**Acknowledgments.** The computational resources for this research were supported by the Accelerating Higher Education Expansion and Development (AHEAD) Operation Grant No. 6026-LK/8743-LK from the Ministry of Higher Education, Sri Lanka, funded by the World Bank and the National Research Council of Sri Lanka Grant No. 19-080.

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

## A  Appendix

### A.1  The Use of Large Language Models (LLMs)

**LLM Usage Statement:** We made limited use of large language models to enhance the clarity and readability of the text. They were not involved in the conception of ideas, experiment design, analysis, or the production of results.

### A.2  Ethics Statement

We confirm that our research adheres to the highest standards of ethical considerations. All work presented in this paper is original, and any external contributions or sources have been appropriately cited. Our study does not introduce new datasets, nor does it involve experiments utilizing demographic or identity characteristics.

### A.3 Reproducibility Statement

To ensure the reproducibility of our work, we have detailed the comprehensive training procedures, hyperparameters, and experimental settings in Sections 4 and A.9 of the paper.

### A.4 Limitations

A notable limitation of the proposed angular diversity-based calibration method is a slight reduction in classification accuracy observed in certain scenarios. Although empirical results suggest that overall accuracy is not significantly affected, some minor degradations in a few cases like semantically overlapping datasets — typically under 1-2% — have been noted in some instances. This trade-off aligns with findings from prior test-time prompt calibration-focused approaches (Yoon et al., 2024; Sharifdeen et al., 2025), which similarly reduce the expected calibration error (ECE) at the expense of minor compromise in accuracy. Such trade-offs are generally considered acceptable, particularly in high-stakes applications where better model calibration, reliability, and trustworthy uncertainty estimation are of critical importance.

Another limitation arises from the method's reliance on test-time adaptation, which limits access to labeled training or validation data, making hyperparameter tuning difficult and limiting optimization during adaptation. Within the A-TPT framework, the trade-off between accuracy and calibration is governed by a regularization hyperparameter $\lambda$ (as defined in Eq. 3), which is fixed at 80.0 for all test instances based on insights from ablation studies. While this fixed value has shown better performance across various settings, we acknowledge that it may not be optimal for each data sample; it serves as a practical starting point. Future research could benefit from exploring dynamic, every data sample adaptation of $\lambda$ to further better calibration performance, without compromising accuracy. Notably, any such method does not depend on labeled data to preserve the real-world applicability of test-time prompt tuning in label-scarce settings.

### A.5 Gradient Analyses Comparison to O-TPT

We analyze and compare the gradients of loss functions for A-TPT with O-TPT. To simplify the derivation, we only consider the norm of the gradient of the objective function, composing the loss function, w.r.t. the corresponding text feature matrix $\mathbf{E}$. For intuitive comparison, the analysis results are presented in Fig. 7.

Corresponding to the O-TPT, the gradient norm is derived as follows:

$$\left\| \frac{\partial \,\hat{\mathbf{E}}\hat{\mathbf{E}}^T}{\partial \mathbf{e}_i} \right\| = \left\| \frac{\partial \left( \frac{\mathbf{e}_i^T \mathbf{e}_j}{\|\mathbf{e}_i\|\|\mathbf{e}_j\|} \right)}{\partial \mathbf{e}_i} \right\| = \frac{\|(\boldsymbol{I} - \boldsymbol{M}_{\mathbf{e}_i})\,\mathbf{e}_j\|}{\|\mathbf{e}_i\|\,\|\mathbf{e}_j\|} = \frac{\|\mathbf{e}_j\|\,\|\sin\boldsymbol{\theta}_{ij}\|}{\|\mathbf{e}_i\|\,\|\mathbf{e}_j\|} = \frac{\|\sin\boldsymbol{\theta}_{ij}\|}{\|\mathbf{e}_i\|}, \quad M_{\mathbf{e}_i} = \frac{\mathbf{e}_i \mathbf{e}_i^T}{\|\mathbf{e}_i\|^2} \tag{3}$$

where $\boldsymbol{M}_{\mathbf{e}_i}$ represents the projection matrix of $\mathbf{e}_i$. From the above derivation and Fig. 7, we can see that the gradient norm is very small when the pairwise angle is close to zero. That is why the orthogonality constraint is hard to converge for the case that $\mathbf{e}_i$ and $\mathbf{e}_j$ are close to each other in both $N < |D|$ & $N > |D|$ categories. Next, we derive the gradient norm corresponding to the A-TPT:

$$\left\| \frac{\partial \boldsymbol{\theta}_{ij}}{\partial \mathbf{e}_i} \right\| = \left\| \frac{\partial \boldsymbol{\theta}_{ij}}{\partial \,\hat{\mathbf{E}}\hat{\mathbf{E}}^T} \frac{\partial \,\hat{\mathbf{E}}\hat{\mathbf{E}}^T}{\partial \mathbf{e}_i} \right\| = \frac{1}{\sin\boldsymbol{\theta}_{ij}} \frac{\sin\boldsymbol{\theta}_{ij}}{\|\mathbf{e}_i\|} = \frac{1}{\|\mathbf{e}_i\|} \tag{4}$$

Compared to the gradient norm corresponding to the orthogonality constraints, as referred to Equation 3, the gradient norm corresponding to the angular diversity is independent of the pairwise angle $\boldsymbol{\theta}_{ij}$, so it would not encounter the very small gradient even though $\boldsymbol{\theta}_{ij}$ is zero. Fig. 7 shows that the gradient norm corresponding to angular diversity empirically proves that the A-TPT is better than the O-TPT, primarily due to its more stable and consistent gradient during test-time prompt tuning. Therefore, directly optimizing angular distance rather than using cosine similarity, and getting near-optimal solutions for the Tammes problem, sufficiently justifies the design choice for both $N < |D|$ & $N > |D|$ categories (Fig. 4), especially for $N > |D|$.

### A.6    COMPUTATIONAL COMPLEXITY

As shown in the Tab. 8, we compare C-TPT, O-TPT, and A-TPT on the larger, higher-dimensional embeddings natural distribution shift dataset (Recht et al., 2019) for the $N > |D|$ case from the perspective of asymptotic complexity, calculating time per batch, occupied memory, and ECE. Compared to L2 Regularization, the orthogonality constraints and angular diversity slightly increase the asymptotic complexity, calculation time, and occupied memory, while reducing ECE.

However, the orthogonality constraints greatly increase that due to the computational overhead of all the pairwise matrix operations. In terms of ECE, the angular diversity reduces over the L2 regularization by a substantial margin. The angular diversity is also the most effective to enlarge the minimal pairwise angular distance of textual features to sharpening class boundaries, which would increase the inter-class separability and better calibration performance. Besides, as a simple plug-in regularizer with negligible computational overhead, it is shown to be pre-trained backbone-agnostic and produces better calibration on fine-grained classification tasks and natural distribution shifts. Therefore, the key advantage of angular diversity highlighted in this paper is not only significant but also scalable.

| Method | Regularization | Asymptotic Complexity | Time (s) /Batch | Memory (MiB) | ECE |
|---|---|---|---|---|---|
| C-TPT | L2 Regularization | $\mathcal{O}(N \cdot |D|)$ | 1.055 | 21838 | 6.23 |
| O-TPT | Orthogonality Constraints | $\mathcal{O}(N^2 \cdot |D|)$ | 1.064 | 23740 | 3.97 |
| **A-TPT (Ours)** | Angular Diversity | $\mathcal{O}(N^2 \cdot |D|)$ | 1.058 | 21840 | **2.96** |

Table 8: Comparison of different regularization methods on ImageNet-V2 (Recht et al., 2019). The angular diversity achieves lower ECE with negligible computational overhead.

### A.7    DETAILS ON THE DATASETS

We evaluate our approach on multiple datasets encompassing both fine-grained classification and natural distribution shift scenarios (suppl. carries details). For fine-grained classification, we conduct experiments using the ImageNet (Deng et al., 2009) and Caltech101 (Fei-Fei et al., 2004) along with diverse datasets across various domains: DTD (Cimpoi et al., 2014) for texture identification, Flower102 (Nilsback & Zisserman, 2008) and OxfordPets (Parkhi et al., 2012) for plants and animal categories, Food101 (Bossard et al., 2014) for food classification, StanfordCars (Krause et al., 2013) and Aircraft (Maji et al., 2013) for transportation classification, SUN397 (Xiao et al., 2010) for scene categorization, UCF101 (Soomro et al., 2012) for human action recognition, and EuroSAT (Helber et al., 2018) for satellite imagery in environmental categorization. For natural distribution shifts, we utilize several ImageNet variants: ImageNet-V2 (Recht et al., 2019) (natural images), ImageNet-A (Hendrycks et al., 2021b) (natural adversarial examples), ImageNet-R (Hendrycks et al., 2021a) (artistic renditions), and ImageNet-Sketch (Wang et al., 2019) (black and white sketches) datasets, as benchmarks for out-of-distribution (OOD) performance evaluation.

Tab. 9 (left) summarizes the details of each dataset, such as the number of classes, test set size, and the corresponding task descriptions.

### A.8    CLIP'S EMBEDDING DIMENSION

Tab. 10 (right) presents the embedding dimensions associated with various CLIP backbone architectures. For the ResNet-based models, CLIP RN50 and CLIP RN101 produce 1024-dimensional and 512-dimensional embeddings, respectively. In contrast, the Vision Transformer (ViT) variants, including ViT-B/16, ViT-B/32, and ViT-L/14, generate embeddings of 512, 512, and 768 dimensions, respectively. These embedding sizes reflect the representational capacity of the model and play a crucial role in angular diversity for zero-shot performance, across diverse tasks where the relationship between the number of classes $N$ and the embedding dimension $|D|$ influences calibration ($N < |D|$ and $N > |D|$) via feature angular separation.

| Dataset | # Classes | Test set size | Text feature matrix (**E**) | |
|---|---|---|---|---|
| | | | CLIP ViT-B/16 | CLIP RN50 |
| fine-grained classification datasets | | | $[N, |D|]$ | |
| ImageNet | 1000 | 50,000 | $[1000, 512]$ | $[1000, 1024]$ |
| Caltech101 | 100 | 2,465 | $[100, 512]$ | $[100, 1024]$ |
| OxfordPets | 37 | 3,669 | $[37, 512]$ | $[37, 1024]$ |
| StanfordCars | 196 | 8,041 | $[196, 512]$ | $[196, 1024]$ |
| Flowers102 | 102 | 2,463 | $[102, 512]$ | $[102, 1024]$ |
| Food101 | 101 | 30,300 | $[101, 512]$ | $[101, 1024]$ |
| FGVCAircraft | 100 | 3,333 | $[100, 512]$ | $[100, 1024]$ |
| SUN397 | 397 | 19,850 | $[397, 512]$ | $[397, 1024]$ |
| DTD | 47 | 1,692 | $[47, 512]$ | $[47, 1024]$ |
| EuroSAT | 10 | 8,100 | $[10, 512]$ | $[10, 1024]$ |
| UCF101 | 101 | 3,783 | $[101, 512]$ | $[101, 1024]$ |
| natural distribution shift datasets | | | | |
| ImageNet-A | 200 | 7,500 | $[200, 512]$ | $[200, 1024]$ |
| ImageNetV2 | 1000 | 10,000 | $[1000, 512]$ | $[1000, 1024]$ |
| ImageNet-R | 200 | 30,000 | $[200, 512]$ | $[200, 1024]$ |
| ImageNet-Sketch | 1000 | 50,889 | $[1000, 512]$ | $[1000, 1024]$ |

Table 9: The detailed statistics of datasets used in the experiments.

| CLIP pre-trained backbone | Embedding dimension |
|---|---|
| RN50 | 1024-d |
| RN101 | 512-d |
| ViT-B/16 | 512-d |
| ViT-B/32 | 512-d |
| ViT-L/14 | 768-d |

Table 10: CLIP ResNet and ViT's embedding dimension.

### A.9 IMPLEMENTATION DETAILS

We employ two CLIP backbones: CLIP ViT-B/16 and CLIP RN50. For all experimental setups, we use test-time prompt tuning (TPT) (Shu et al., 2022) as the primary objective to maximize accuracy while incorporating A-TPT as an auxiliary objective to better the calibration performance as described in Eq. 2. We fix the $\lambda$ as 80.0 for all cases unless otherwise specified. We perform prompt optimization for a single-step update with the `AdamW` (Loshchilov & Hutter, 2017) optimizer and set the learning rate to 5e-3. We initialize the prompt embeddings with hard prompts, following C-TPT (Yoon et al., 2024) and all other settings following the standard TPT (Shu et al., 2022) configurations. We conduct all experiments with a batch size of 64 on a single NVIDIA Quadro RTX 6000 GPU (24GB memory).

### A.10 WEIGHTED AVERAGE COMPARISON

The formula for the weighted average metric is:

$$\text{Weighted Average Metric} = \frac{\sum(\text{Test Set Size}_i \times \text{Metric}_i)}{\sum(\text{Test Set Size}_i)} \quad (5)$$

In Tab. 11, we present the weighted average accuracy and ECE results of the proposed A-TPT method based on the test set size, and we observe better calibration performance with both CLIP ViT-B/16 and CLIP RN50 backbones. Our method (A-TPT) significantly outperforms C-TPT as well as O-TPT in terms of calibration performance, resulting in a lower ECE without compromising accuracy. Specifically, with the CLIP ViT-B/16 backbone, the average ECE drops from 4.74 ( C-TPT) and 3.91 (O-TPT) to **2.73** with A-TPT. Similarly, for the CLIP RN50 backbone, ECE reduces from 6.11 (C-TPT) and 4.88 (O-TPT) to **3.32** and outperforms C-TPT and O-TPT with a substantial improvement in both settings.

### A.11 EXPANDED STUDY

We have expanded this study by collecting another 6 prompt templates within 80 different hard prompt styles (Radford et al., 2021). For illustration, consider the following examples from the natural distribution shift ImageNet dataset (Deng et al., 2009) using CLIP ViT-B/16, categorized into well-calibrated and poor-calibrated prompts:

| Method | Metric | CLIP ViT-B/16 | CLIP RN50 |
|---|---|---|---|
| Baseline | Acc. | 62.99 | 51.75 |
| | ECE | 3.93 | 4.04 |
| TPT | Acc. | 64.84 | 54.00 |
| | ECE | 10.23 | 11.48 |
| C-TPT | Acc. | 64.60 | 53.64 |
| | ECE | 4.74 | 6.11 |
| O-TPT | Acc. | 63.54 | 52.51 |
| | ECE | 3.91 | 4.88 |
| **A-TPT (Ours)** | **Acc.** | 63.89 | 52.40 |
| | **ECE** | **2.73** | **3.32** |

Table 11: Comparison of Weighted Average Accuracy and Expected Calibration Error (ECE) with CLIP ViT-B/16 and CLIP RN50 backbones.

- well-calibrated prompts (similar accuracy, higher AD, lower ECE):
  - `a photo of a [CLS]` - Acc: 66.8, ECE: 2.12, AD: 0.65
  - `a good photo of the [CLS]` - Acc: 66.7, ECE: 3.46, AD: 0.62
  - `a photo of the weird [CLS]` - Acc: 67.1, ECE: 5.38, AD: 0.57

- poor-calibrated prompts (lower AD, higher ECE):
  - `a sculpture of a [CLS]` - Acc: 61.8, ECE: 6.08, AD: 0.54
  - `graffiti of a [CLS]` - Acc: 62.4, ECE: 4.19, AD: 0.57
  - `a cartoon [CLS]` - Acc: 62.1, ECE: 2.24, AD: 0.59

As shown, these examples provide important insight into the relationship between AD and calibration error (ECE) within the same accuracy group, aligning with the empirical findings of our paper. We agree that it needs a lot of prompt templates, datasets, and models to support the conclusion of Figure 4 further. However, finding prompt templates within the same accuracy group across different hard prompt styles (Radford et al., 2021) is a challenging task.

Lastly, while different prompt template certainly affects calibration error, our empirical results suggest that accuracy is not a sufficient indicator of calibration, and directly optimizing for angular diversity leads to improved ECE across different settings.

## A.12 RESULTS ON STANDARD DEVIATION ACROSS RANDOM SEEDS

Tab. 12 presents the mean and standard deviation of accuracy and ECE over three random runs with different seeds for A-TPT across five datasets with CLIP ViT-B/16 backbone. Compared to C-TPT and O-TPT, A-TPT shows a lower mean standard deviation in both accuracy and ECE, indicating more stable performance to randomness in prompt initialization and greater consistency in calibration. This consistency highlights A-TPT's reliability in maintaining stable performance across runs, a critical property for practical deployment in scenarios that demand reproducibility and calibration stability.

| Method | Metric | DTD | Flowers102 | Food101 | Caltech101 | Standford Cars | Average |
|---|---|---|---|---|---|---|---|
| **Pre-trained Backbone: CLIP ViT-B/16** | *Embedding dimension: 512-d* | | | | | | |
| C-TPT | Std. Acc. | ±.12 | ±.16 | ±.16 | ±.22 | ±.20 | ±.17 |
| | Std. ECE | ±.24 | ±.18 | ±.24 | ±.12 | ±.19 | ±.19 |
| O-TPT | Std. Acc. | ±.14 | ±.11 | ±.03 | ±.10 | ±.19 | ±.11 |
| | Std. ECE | ±.17 | ±.25 | ±.14 | ±.20 | ±.10 | ±.18 |
| **A-TPT (Ours)** | **Std. Acc.** | ±.15 | ±.09 | ±.04 | ±.08 | ±.14 | ±.10 |
| | **Std. ECE** | ±.12 | ±.15 | ±.10 | ±.09 | ±.08 | ±.0.11 |

Table 12: Standard deviation of three random runs with different seeds with CLIP ViT-B/16 backbone.

## A.13 RESULTS WITH OTHER CALIBRATION METRICS: SCE CALIBRATION PERFORMANCE COMPARISON

Since the ECE metric could suffer from bias, In addition to evaluating model calibration through Expected Calibration Error (ECE), we also examine calibration using Static Calibration Error (SCE) (Nixon et al., 2019), which serves as a class-wise variant of ECE. Tab. 13 compares the SCE results across ten fine-grained classification datasets with the CLIP ViT-B/16 and CLIP RN50 backbones. The

performance of our proposed A-TPT method is benchmarked against several alternative approaches, including the Baseline, TPT, C-TPT, and O-TPT. A-TPT consistently achieves superior calibration performance, demonstrating a substantial reduction in SCE across all datasets. For the CLIP-ViT-B/16 backbone, our method outperforms the alternatives, with an overall average SCE reduction of up to **0.89** compared to 1.06 for Baseline, 1.15 for TPT, 1.11 for C-TPT, and 1.07 for O-TPT. For the CLIP RN50 **1.03** compared to 1.22 for Baseline, 1.30 for TPT, 1.27 for C-TPT, and 1.24 for O-TPT. This substantial improvement highlights the efficacy of angular diversity in better prompt calibration performance.

| Method | DTD | Flowers102 | Food101 | SUN397 | Aircrafts | OxfordPets | Caltech101 | UCF101 | EuroSAT | Standford Cars | Average |
|---|---|---|---|---|---|---|---|---|---|---|---|
| **Pre-trained Backbone: CLIP ViT-B/16** | **Metric: SCE** | *Embedding dimension: 512-d* | | | | | | | | | |
| Baseline | 1.33 | 0.59 | 0.20 | 0.12 | 0.52 | 0.68 | 0.25 | 0.52 | 6.18 | 0.23 | 1.06 |
| TPT | 1.44 | 0.51 | 0.17 | 0.15 | 0.58 | 0.60 | 0.16 | 0.57 | 7.07 | 0.25 | 1.15 |
| C-TPT | 1.31 | 0.52 | 0.22 | 0.14 | 0.56 | 0.58 | 0.22 | 0.52 | 6.81 | 0.22 | 1.11 |
| O-TPT | 1.24 | 0.53 | 0.19 | 0.12 | 0.56 | 0.57 | 0.17 | 0.51 | 6.58 | 1.07 | 1.07 |
| **A-TPT** | 1.11 | 0.51 | 0.15 | 0.11 | 0.53 | 0.56 | 0.13 | 0.49 | 5.04 | 0.22 | **0.89** |
| **Pre-trained Backbone: CLIP RN50** | **Metric: SCE** | *Embedding dimension: 1024-d* | | | | | | | | | |
| Baseline | 1.31 | 0.66 | 0.29 | 0.12 | 0.54 | 0.73 | 0.35 | 0.54 | 7.39 | 0.23 | 1.22 |
| TPT | 1.52 | 0.63 | 0.25 | 0.11 | 0.60 | 0.54 | 0.38 | 0.51 | 8.23 | 0.24 | 1.30 |
| C-TPT | 1.43 | 0.62 | 0.26 | 0.11 | 0.53 | 0.67 | 0.32 | 0.51 | 8.07 | 0.23 | 1.27 |
| O-TPT | 1.34 | 0.60 | 0.27 | 0.12 | 0.51 | 0.69 | 0.30 | 0.50 | 7.85 | 0.22 | 1.24 |
| **A-TPT** | 1.16 | 0.59 | 0.24 | 0.10 | 0.49 | 0.58 | 0.28 | 0.49 | 6.20 | 0.21 | **1.03** |

Table 13: Static Calibration Error (SCE) (10e-2) performance comparison across fine-grained datasets with CLIP ViT-B/16 and CLIP RN50 backbone. The overall best-performing result is in bold.

### A.14    RELIABILITY PLOTS AND CONFIDENCE HISTOGRAM

Figs. 8 and 9 present the reliability diagrams for the CLIP ViT-B/16 and CLIP RN50 backbones, respectively, comparing the calibration performance of C-TPT, O-TPT, and A-TPT across the Aircraft, UCF101, StandfordCars, and SUN397 datasets. For the CLIP ViT-B/16 backbone (Fig. 8), A-TPT exhibits a marked improvement in addressing the overconfidence problem, outperforming both O-TPT and C-TPT as evidenced in the reliability diagrams presented in the *first*, *second* and *third* rows of Fig. 8. Similarly, the results corresponding to the CLIP RN50 backbone, (Fig. 9) show that A-TPT yields substantially better calibration compared to O-TPT and C-TPT, particularly in reducing the prevalence of overconfident predictions. Additional reliability diagrams for fine-grained classification tasks and natural distribution shifts — evaluated on EuroSAT ($N < |D|$), ImageNet-V2, K ($N > |D|$) datasets with both backbones — are provided in Figs. 10a, 11, further validating the potency of A-TPT under varying conditions.

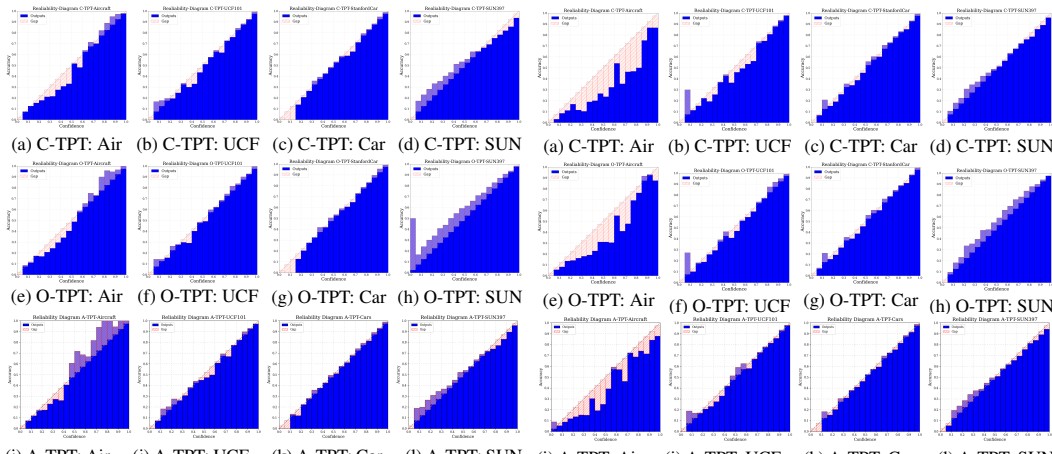

(a) C-TPT: Air    (b) C-TPT: UCF    (c) C-TPT: Car    (d) C-TPT: SUN    (a) C-TPT: Air    (b) C-TPT: UCF    (c) C-TPT: Car    (d) C-TPT: SUN

(e) O-TPT: Air    (f) O-TPT: UCF    (g) O-TPT: Car    (h) O-TPT: SUN    (e) O-TPT: Air    (f) O-TPT: UCF    (g) O-TPT: Car    (h) O-TPT: SUN

(i) A-TPT: Air    (j) A-TPT: UCF    (k) A-TPT: Car    (l) A-TPT: SUN    (i) A-TPT: Air    (j) A-TPT: UCF    (k) A-TPT: Car    (l) A-TPT: SUN

Figure 8: Reliability diagrams for CLIP ViT-B/16 backbone.

Figure 9: Reliability diagrams for CLIP RN50 backbone.

In addition to the reliability diagrams on EuroSAT (Fig. 10a), confidence histogram diagrams (as shown in Fig. 10b) are also included to provide a complementary perspective on the distribution of

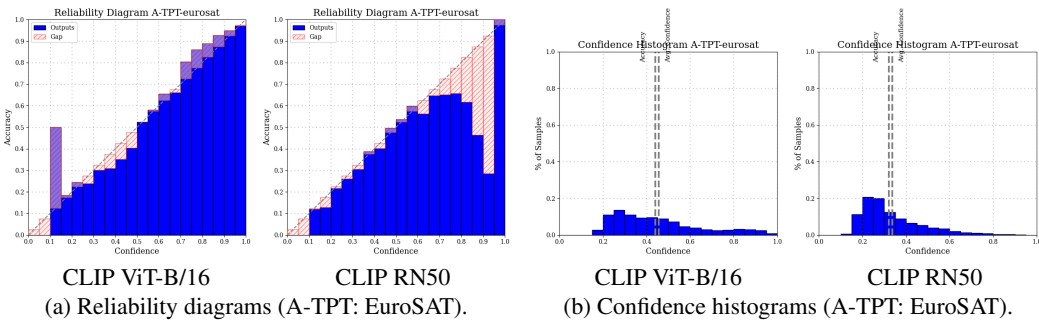

CLIP ViT-B/16     CLIP RN50       CLIP ViT-B/16     CLIP RN50

(a) Reliability diagrams (A-TPT: EuroSAT).    (b) Confidence histograms (A-TPT: EuroSAT).

Figure 10: Reliability and confidence histogram diagrams for EuroSAT on fine-grained classification tasks with CLIP ViT-B/16 ($N < |D|$, *left*) and CLIP RN50 ($N < |D|$, *right*) backbone.

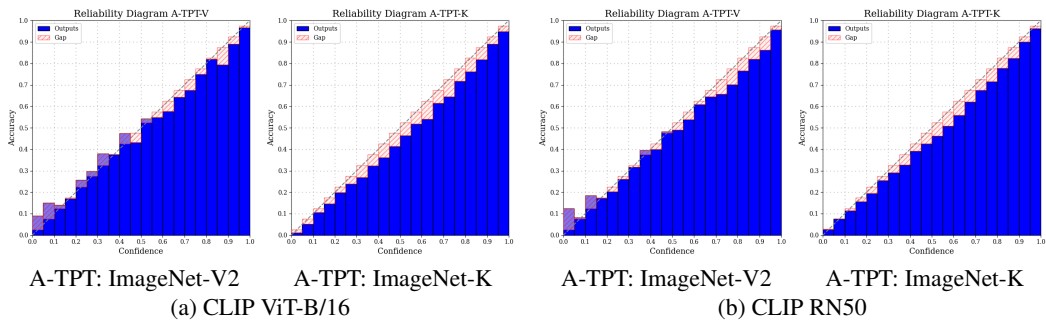

A-TPT: ImageNet-V2    A-TPT: ImageNet-K     A-TPT: ImageNet-V2    A-TPT: ImageNet-K

(a) CLIP ViT-B/16            (b) CLIP RN50

Figure 11: Reliability diagrams for (ImageNet-V2,K) on natural distribution shifts with CLIP ViT-B/16 ($N > |D|$) and CLIP RN50 ($N < |D|$) backbone, respectively.

model confidence scores. These histograms illustrate how frequently different confidence levels are assigned to predictions, thereby offering deeper insight into the calibration characteristics of each method. A well-calibrated model is expected to produce a confidence distribution that aligns closely with the true likelihood of correctness, and the confidence histograms further highlight the extent to which A-TPT mitigates overconfident predictions.

### A.15 WHY HYPERSPHERE OFFERS OPTIMAL TEXTUAL FEATURE SEPARATION IN A-TPT?

Existing test-time prompt tuning techniques (Yoon et al., 2024; Sharifdeen et al., 2025) may not guarantee optimal angular separation among textual features distributed across the hypersphere, which can often result in inconsistent fluctuating, but slightly lower (for $N > |D|$, where angular separation (orthogonalization) fails), and showing higher (for $N < |D|$, underutilize hyperspherical space, which is available to achieve even greater angular separation (even lower cosine similarities) in cases where TPT fails (these challenging points), as illustrated in Fig. 2, cosine similarities between textual features, thereby leading to suboptimal model calibration performance. For better calibration, we hypothesize that mere feature dispersion and orthogonalization are insufficient; instead, promoting uniform angular separation across textual features is more beneficial to achieving optimal calibration performance.

Figs. 12 and 13 present a comparative analysis of the cosine similarity distribution among textual features generated by the O-TPT and our proposed A-TPT method under varying dataset regimes. Fig. 12 examines the natural distribution shifts (ImageNet-V2 (Recht et al., 2019)) with the CLIP ViT-B/16 backbone, which is shown for the case where the number of classes exceeds the embedding dimensionality ($N > |D|$). Under this regime, A-TPT exhibits a slightly higher but markedly more consistent mean cosine similarity score across each data sample relative to O-TPT. The corresponding histograms further illustrate that A-TPT produces a narrower distribution (lower variance) of cosine similarities, indicating greater angular uniformity and stability in the angular relationships among textual feature embeddings.

Fig. 13 present a similar analysis on the fine-grained classification tasks (EuroSAT (Helber et al., 2018)) with the same backbone, representing the scenario where the number of classes is lesser

than the embedding dimensionality ($N < |D|$). In this setting, A-TPT yields a substantially lower and more consistent mean cosine similarity score and narrower distribution (lower variance) around the mean relative to O-TPT, as evidenced by both the sample-wise cosine similarity plots and corresponding histograms. This suggests that A-TPT is particularly effective at promoting greater angular diversity and increasing feature dispersion when the hyperspherical space is underutilized.

Overall, these findings highlight A-TPT's capability to enforce stable and uniform angular separation among textual features, regardless of the relationship between the number of classes and embedding dimensionality. These empirical findings validate that angular diversity enables optimal angular separation, better utilization of the hyperspherical space, and increased dispersion among features are indicative of clearer class boundaries and improved calibration performance.

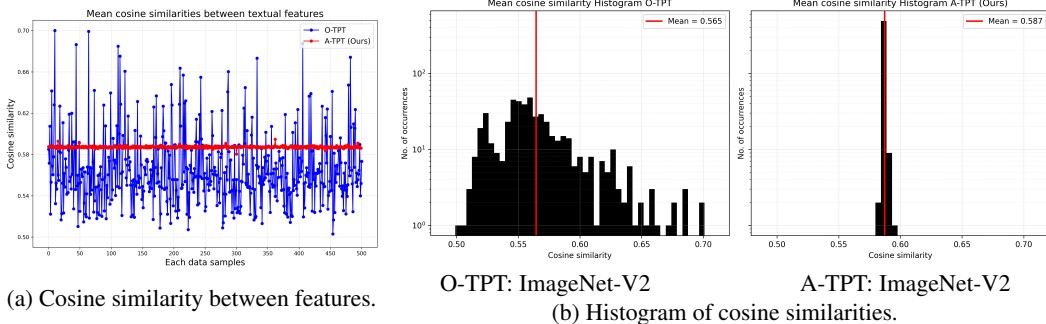

(a) Cosine similarity between features.  (b) Histogram of cosine similarities.

Figure 12: Comparison of mean cosine similarity changes on a natural distribution dataset (ImageNet-V2) (Recht et al., 2019)) with CLIP ViT-B/16 backbone. When $N > |D|$ our A-TPT offers slightly higher but more consistent cosine similarity values among text features for all the data points.

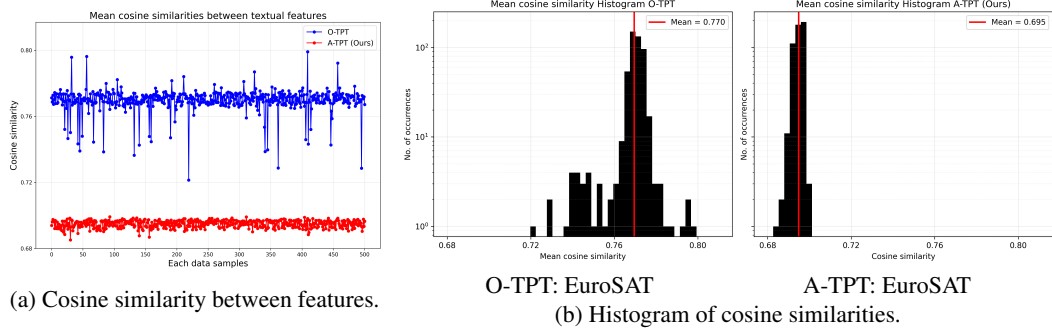

(a) Cosine similarity between features.  (b) Histogram of cosine similarities.

Figure 13: Comparison of mean cosine similarity changes on a fine-grained classification dataset (EuroSAT (Helber et al., 2018)) with CLIP ViT-B/16 backbone. When $N < |D|$ our A-TPT offers much lower and more consistent cosine similarity values among text features for all the data points.

### A.16 SOME THEORETICAL ASPECTS ON A-TPT OVER O-TPT.

Firstly, based on the Tammes problem (Tammes, 1930), the uniform distribution (best-packing) means the distance between two points is maximized; therefore, while A-TPT builds upon the TPT framework, it introduces a novel angular diversity that is boosted to the utmost extent. Angular diversity (AD) focuses on the uniform distribution — maximizing the angular distance between textual features corresponding to their actual class labels, thereby maximizing the inter-class feature separability for better prompt calibration. Prior work (Wang & Isola, 2020) has shown that uniformity (uniformly distributed feature points on the unit hypersphere) preserves maximal information, closely associated with strong zero-shot CLIP performance. Therefore, we argue that uniformly distributed textual features are the way to ensure better class separation and improve VLM calibration. Unlike orthogonality constraints pursue orthogonality for all the pairwise textual features and suffer from poor calibration when $N > |D|$, our numerical optimization (A-TPT) is robust with negligible computational overhead across both $N > |D|$ and $N < |D|$ settings. Secondly, inspired by insights from ArcFace (Deng et al., 2019), Figs. 2, 4, and Appendix A.15 of the main paper demonstrate that AD gets the greatest minimum pairwise angular distance across all $N > |D|$ & $N < |D|$ cases, and therefore the most diverse prompt vectors. Thirdly, as discussed in the paper, orthogonal regularization tends to group textual features closer, especially when the number of classes is greater

| Method | Metric | DTD | Flowers102 | Food101 | SUN397 | Aircrafts | OxfordPets | Caltech101 | UCF101 | EuroSAT | Standford Cars | Average |
|---|---|---|---|---|---|---|---|---|---|---|---|---|
| **Pre-trained Backbone: CLIP ViT-B/16** | *Embedding dimension: 512-d* | | | | | | | | | | | |
| Baseline | Acc. | 38.40 | 64.50 | 81.40 | 62.40 | 22.70 | 86.20 | 88.10 | 67.60 | 34.60 | 66.50 | 61.24 |
| | ECE | 7.43 | 4.59 | 1.10 | 6.11 | 2.83 | 7.43 | 14.10 | 2.65 | 14.10 | 4.59 | 7.01 |
| TPT | Acc. | 45.50 | 67.90 | 84.90 | 65.90 | 24.50 | 87.40 | 91.50 | 66.40 | 43.30 | 67.20 | 64.45 |
| | ECE | 20.00 | 14.60 | 5.74 | 13.30 | 19.20 | 6.34 | 3.11 | 14.10 | 18.20 | 6.36 | 12.09 |
| C-TPT | Acc. | 46.30 | 69.60 | 84.10 | 65.50 | 24.70 | 88.80 | 91.70 | 67.00 | 43.00 | 66.90 | 64.76 |
| | ECE | 18.00 | 10.60 | 2.43 | 10.70 | 10.50 | 1.59 | 1.89 | 7.42 | 8.73 | 1.64 | 7.35 |
| O-TPT | Acc. | 44.62 | 68.29 | 84.82 | 63.05 | 23.16 | 88.28 | 91.48 | 64.74 | 44.81 | 66.02 | 63.92 |
| | ECE | 12.85 | 4.67 | 1.85 | 2.67 | 6.37 | 3.59 | 3.00 | 4.08 | 8.33 | 2.71 | 5.01 |
| **A-TPT** | **Acc.** | 40.84 | 69.27 | 82.84 | 62.59 | 23.58 | 87.54 | 92.13 | 67.38 | 43.90 | 66.54 | 63.66 |
| | **ECE** | 6.91 | 2.95 | 1.38 | 2.17 | 4.24 | 2.22 | 2.28 | 3.47 | 2.47 | 2.45 | **3.05** |
| **Pre-trained Backbone: CLIP RN50** | *Embedding dimension: 1024-d* | | | | | | | | | | | |
| Baseline | Acc. | 39.60 | 57.70 | 73.00 | 56.50 | 16.10 | 79.80 | 80.90 | 56.30 | 21.90 | 56.90 | 60.24 |
| | ECE | 6.94 | 5.14 | 1.49 | 3.33 | 6.42 | 3.30 | 4.79 | 3.76 | 13.90 | 4.83 | 5.39 |
| TPT | Acc. | 39.20 | 61.60 | 75.80 | 60.20 | 17.40 | 82.60 | 86.50 | 59.70 | 26.30 | 58.80 | 56.81 |
| | ECE | 24.80 | 17.00 | 7.93 | 11.40 | 17.50 | 7.31 | 6.02 | 14.40 | 15.70 | 4.49 | 12.65 |
| C-TPT | Acc. | 39.10 | 67.00 | 76.00 | 60.30 | 17.40 | 83.50 | 87.10 | 59.60 | 26.10 | 57.20 | 57.33 |
| | ECE | 18.00 | 6.34 | 3.70 | 8.28 | 13.50 | 1.75 | 2.85 | 8.82 | 11.20 | 1.65 | 7.61 |
| O-TPT | Acc. | 40.54 | 65.49 | 75.51 | 58.98 | 15.99 | 83.78 | 86.98 | 58.79 | 26.89 | 56.77 | 56.97 |
| | ECE | 12.42 | 3.03 | 1.32 | 3.35 | 8.36 | 4.47 | 3.53 | 3.27 | 7.21 | 2.74 | 4.97 |
| **A-TPT** | **Acc.** | 39.79 | 64.32 | 74.21 | 58.36 | 16.03 | 83.44 | 86.89 | 58.33 | 26.37 | 56.37 | 56.41 |
| | **ECE** | 6.61 | 2.97 | 1.39 | 2.79 | 6.11 | 3.53 | 2.89 | 2.83 | 3.34 | 2.19 | **3.46** |

Table 14: Comparison of calibration performance with CLIP ViT-B/16 and CLIP RN50 backbone with the prompt of "`a photo of the cool [CLS]`".

| Method | Metric | DTD | Flowers102 | Food101 | SUN397 | Aircrafts | OxfordPets | Caltech101 | UCF101 | EuroSAT | Standford Cars | Average |
|---|---|---|---|---|---|---|---|---|---|---|---|---|
| **Pre-trained Backbone: CLIP ViT-B/16** | *Embedding dimension: 512-d* | | | | | | | | | | | |
| Baseline | Acc. | 42.40 | 64.70 | 83.90 | 61.40 | 22.30 | 82.50 | 90.90 | 64.80 | 38.80 | 64.60 | 61.63 |
| | ECE | 4.94 | 4.70 | 2.78 | 3.33 | 7.09 | 2.91 | 7.51 | 2.79 | 13.40 | 2.49 | 5.64 |
| TPT | Acc. | 45.80 | 69.40 | 84.80 | 65.30 | 22.90 | 83.00 | 93.00 | 67.10 | 40.70 | 67.30 | 63.93 |
| | ECE | 20.50 | 12.20 | 5.05 | 7.94 | 16.20 | 7.30 | 2.91 | 11.60 | 20.80 | 6.26 | 11.07 |
| C-TPT | Acc. | 45.40 | 71.50 | 84.30 | 66.00 | 23.60 | 86.90 | 93.80 | 66.40 | 51.50 | 66.60 | 65.60 |
| | ECE | 15.50 | 4.49 | 1.36 | 3.54 | 9.05 | 2.89 | 1.62 | 3.87 | 5.18 | 1.75 | 4.93 |
| O-TPT | Acc. | 45.45 | 70.32 | 84.79 | 64.50 | 22.77 | 87.76 | 93.35 | 65.40 | 51.01 | 66.25 | 65.16 |
| | ECE | 11.79 | 3.22 | 2.92 | 4.62 | 7.92 | 3.29 | 3.24 | 2.63 | 5.08 | 1.92 | 4.66 |
| **A-TPT** | **Acc.** | 43.44 | 70.53 | 84.80 | 65.37 | 22.42 | 86.64 | 93.32 | 65.50 | 46.10 | 65.12 | 64.32 |
| | **ECE** | 6.87 | 3.13 | 1.64 | 3.44 | 6.15 | 2.44 | 2.31 | 2.54 | 3.20 | 1.31 | **3.30** |
| **Pre-trained Backbone: CLIP RN50** | *Embedding dimension: 1024-d* | | | | | | | | | | | |
| Baseline | Acc. | 41.10 | 58.10 | 75.20 | 56.20 | 16.10 | 75.70 | 80.30 | 56.30 | 25.50 | 55.80 | 48.45 |
| | ECE | 5.20 | 3.04 | 3.31 | 3.68 | 4.80 | 2.52 | 7.91 | 3.76 | 9.43 | 4.80 | 4.85 |
| TPT | Acc. | 41.20 | 62.70 | 76.10 | 60.70 | 17.90 | 77.20 | 87.10 | 57.70 | 29.40 | 57.70 | 56.77 |
| | ECE | 20.20 | 12.20 | 4.83 | 8.19 | 15.20 | 6.98 | 5.12 | 15.30 | 11.10 | 5.52 | 10.46 |
| C-TPT | Acc. | 41.20 | 65.40 | 75.80 | 61.40 | 17.60 | 78.00 | 88.40 | 58.40 | 30.40 | 57.10 | 57.37 |
| | ECE | 15.60 | 2.97 | 1.90 | 4.84 | 7.16 | 2.72 | 2.89 | 6.99 | 7.69 | 2.05 | 5.48 |
| O-TPT | Acc. | 41.19 | 65.49 | 75.62 | 60.97 | 16.71 | 77.79 | 88.36 | 57.94 | 33.32 | 56.73 | 57.41 |
| | ECE | 13.59 | 2.49 | 1.47 | 3.38 | 6.60 | 2.55 | 2.56 | 6.20 | 5.07 | 2.69 | 4.66 |
| **A-TPT** | **Acc.** | 41.09 | 65.24 | 75.36 | 59.85 | 16.43 | 77.64 | 87.70 | 58.64 | 31.82 | 56.51 | 57.03 |
| | **ECE** | 7.05 | 2.43 | 1.22 | 3.19 | 6.15 | 2.52 | 2.18 | 5.57 | 2.76 | 2.52 | **3.56** |

Table 15: Comparison of calibration performance with CLIP ViT-B/16 and CLIP RN50 backbone with the prompt of "`an example of [CLS]`".

than the embedding dimension. Fourthly, we argue that the improvement over other methods is significant.

## A.17  CALIBRATION PERFORMANCE OF DIFFERENT PROMPT INITIALIZATIONS

This section evaluates the calibration performance of the proposed A-TPT approach when initialized with different prompt templates, such as "`a photo of the cool [CLS]`" and "`an example of [CLS]`", evaluated across CLIP ViT-B/16 and CLIP RN50 backbones. Tab. 14 reports the calibration results of A-TPT with the prompt "`a photo of the cool [CLS]`" with 5 tunable context tokens. For CLIP ViT-B/16 backbone, A-TPT achieves an overall reduced Expected Calibration Error (ECE) of **3.05**, compared to 5.01 for O-TPT and 7.35 for C-TPT. Similarly, with

the CLIP RN50 backbone, attains a calibration error of **3.46**, compared to 4.97 for O-TPT and 7.61 for C-TPT.

Similarly, Tab. 15 presents the calibration results for the prompt "`an example of [CLS]`" with 4 tunable context tokens. In this setting, A-TPT again outperforms C-TPT, achieving a reduced calibration error of **3.30** (CLIP ViT-B/16) compared to 4.66 for O-TPT, and 4.93 for C-TPT. For (CLIP RN50) backbone, A-TPT attains an ECE of **3.56** compared to 4.66 for O-TPT and 5.48 for C-TPT. These results consistently demonstrate that A-TPT maintains strong calibration capabilities (reduces calibration errors) across different prompt initializations, showcasing its robustness and adaptability in diverse settings and affirming its potency in better prompt-based VLM calibration.

### A.18 CALIBRATION PERFORMANCE WITH COMBINED C-TPT AND A-TPT

Tab. 16 presents the calibration results of a combined approach that integrates C-TPT and A-TPT. The findings indicate that combining A-TPT with C-TPT leads to superior calibration performance and can outcompete A-TPT alone. This enhancement reveals the generalizability of A-TPT over a stronger baseline.

| Method | Metric | DTD | Flowers102 | UCF101 |
|---|---|---|---|---|
| C-TPT | Acc. | 46.00 | 69.80 | 65.70 |
| | ECE | 11.90 | 5.04 | 2.54 |
| A-TPT | Acc. | 45.51 | 69.22 | 66.16 |
| | ECE | 4.76 | 3.61 | 2.12 |
| **C-TPT + A-TPT** | **Acc.** | 45.15 | 69.51 | 66.23 |
| | **ECE** | **4.11** | **3.49** | **1.96** |

Table 16: C-TPT + A-TPT on DTD, Flowers102 and UCF101.

### A.19 PARETO FRONT: VISUALIZING THE EFFECT OF A-TPT

Fig. 14 presents the Pareto frontier analysis on the Flowers102 and Food101 datasets, highlighting the trade-off between classification accuracy and Expected Calibration Error (ECE) across varying values of $\lambda$'s. The proposed A-TPT method does not merely achieve higher ECE at the expense of lower accuracy. Instead, it seeks an optimal balance between these two metrics than TPT, C-TPT, and O-TPT across a wide range of $\lambda$ settings in two datasets. This suggests that A-TPT effectively optimizes the trade-off, achieving superior model calibration performance without compromising predictive accuracy and providing a more detailed picture of the performance characteristics of A-TPT, ensuring that the increased performance is not solely due to the better hyperparameter $\lambda$.

The difference in calibration behaviour of the Flowers102 and Food101 datasets when increasing $\lambda$ is due to dataset characteristics and how model confidence (over- vs. under-confidence) is distributed. Refer to Fig. 5, the reliability diagrams of C-TPT and O-TPT show overconfidence on Flowers102 and underconfidence on Food101.

- Flowers102: likely contains more fine-grained, visually similar classes, where the baseline model often makes incorrect predictions with high confidence (overconfidence). To correct overconfidence, the model must lower its confidence in ambiguous samples. This necessary softening of the probability distribution can occasionally shift a high-confidence incorrect prediction to a low-confidence correct or incorrect state. This results in the observed trade-off with increased trading, some accuracy as the model becomes more conservative to improve ECE.

- Food101: features more visually distinct classes, where the model makes correct predictions with uncertain or unreliable confidence estimates (under-confidence). Increasing maximizes the margin between these distinct features. This pushes the correct class probabilities higher (mitigating under-confidence) without introducing ambiguity. This helps A-TPT to lower ECE without sacrificing accuracy (or even gaining a little by sharpening the decision boundary).

That's why A-TPT shows a trend trading some accuracy for improved ECE on Flowers102, while on Food101, A-TPT shows a contrasting pattern, lower ECE without sacrificing accuracy (or even gaining a little) for better calibration.

As shown in the Fig. 14, A-TPT seeks a better balance overall rather than sacrificing accuracy for better calibration, exactly as observed on Food101, while the accuracy-ECE trade-off on Flower102 across $\lambda$, confirms that increased performance could not be a result of a better $\lambda$.

The reported curves are averaged over three seed runs. In Tab. 12, we have reported 3-seed variability for A-TPT (ViT-B/16), with low standard deviations (Flowers102: Std Acc. $\pm 0.09$ Std ECE $\pm 0.15$; Food101: Std Acc. $\pm 0.04$, Std ECE $\pm 0.10$), confirming the curve shapes are stable across runs.

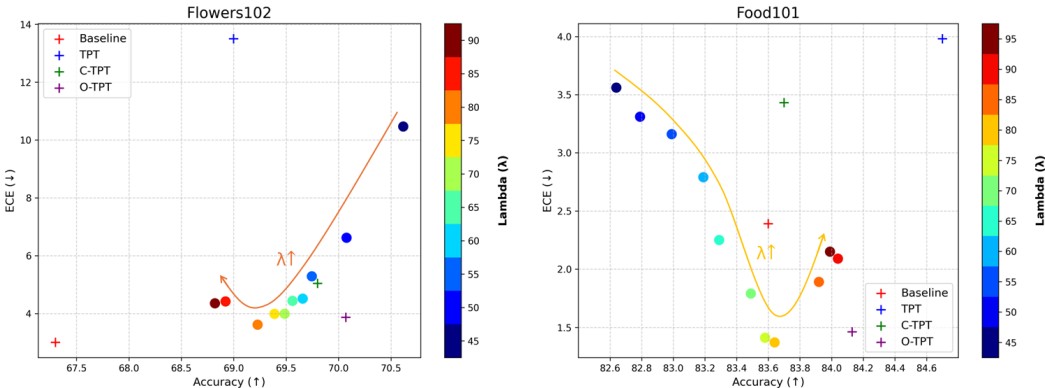

Figure 14: Pareto front analysis on Flowers102 and Food101. The ○ represents A-TPT (Ours), compared against zero-shot baseline, TPT, C-TPT, and O-TPT.

### A.20 CAN ANGULAR DIVERSITY LOSS DEGRADE ACCURACY FOR SEMANTICALLY OVERLAPPING CLASSES?

We distinguish A-TPT results on fine-grained, semantically overlapping datasets like Oxford Pets (37 breeds), Stanford Cars (196 models), FGVC Aircraft (100 variants), Flowers-102, and Food-101 across CLIP-ViT-B/16 and RN50 backbone.

| Method | Metric | OxfordPets | Stanford Cars | Aircrafts | Flowers102 | Food101 | Overall |
|---|---|---|---|---|---|---|---|
| **Pre-trained Backbone: CLIP ViT-B/16** | *Embedding dimension: 512-d* | | | | | | |
| TPT | Acc. | 87.10 | 66.30 | 23.40 | 69.00 | 84.70 | 66.10 |
| | ECE | 5.77 | 5.16 | 16.80 | 13.50 | 3.98 | 9.04 |
| C-TPT | Acc. | 88.20 | 65.80 | 24.85 | 69.80 | 83.70 | 66.47 |
| | ECE | 1.90 | 1.59 | 4.36 | 5.04 | 3.43 | 3.26 |
| O-TPT | Acc. | 87.95 | 64.53 | 23.64 | 70.07 | 84.13 | 66.06 |
| | ECE | 1.90 | 1.78 | 3.68 | 3.87 | 1.46 | 2.54 |
| **A-TPT (Ours)** | **Acc.** | 88.33 | 65.78 | 23.76 | 69.22 | 83.64 | 66.15 |
| | **ECE** | 1.17 | 1.09 | 3.14 | 3.61 | 1.37 | **2.08** |
| **Pre-trained Backbone: CLIP RN50** | *Embedding dimension: 1024-d* | | | | | | |
| TPT | Acc. | 84.50 | 58.00 | 17.00 | 62.50 | 74.90 | 59.38 |
| | ECE | 3.65 | 3.76 | 16.10 | 13.40 | 5.25 | 8.43 |
| C-TPT | Acc. | 84.10 | 56.50 | 17.00 | 65.20 | 74.70 | 59.50 |
| | ECE | 2.77 | 1.94 | 10.70 | 4.14 | 1.86 | 4.28 |
| O-TPT | Acc. | 83.40 | 56.44 | 16.77 | 65.61 | 74.22 | 59.29 |
| | ECE | 3.50 | 1.69 | 8.18 | 2.50 | 1.20 | 3.41 |
| A-TPT (Ours) | **Acc.** | 83.48 | 57.08 | 14.58 | 64.89 | 74.10 | 58.83 |
| | **ECE** | 2.47 | 1.38 | 6.14 | 2.39 | 1.11 | **2.70** |

Table 17: Accuracy and ECE on semantically overlapping datasets. A-TPT preserve accuracy comparable to C-TPT/O-TPT while consistently lowering ECE.

- CLIP VIT-B/16: Overall Acc. remains stable ($66.10 \rightarrow 66.15$), while ECE drops significantly ($9.04 \rightarrow 2.08$). Notably, accuracy actually improves on datasets like OxfordPets ($87.10 \rightarrow 88.33$) and Aircrafts ($23.40 \rightarrow 23.76$)
- CLIP RN50: We observe a minor trade-off in overall accuracy ($59.38 \rightarrow 58.83$), while ECE drops significantly ($8.43 \rightarrow 2.70$). Notably, accuracy actually improves on datasets like Flowers102 ($62.50 \rightarrow 64.89$).

Unlike orthogonality constraints (O-TPT) that force features to be 90° apart, potentially breaking semantic clusters, A-TPT solves the Tammes best-packing problem. It maximizes the minimum angular distance available. This allows semantically overlapping classes to remain relatively closer to each other than to dissimilar classes, provided they utilize the hypersphere effectively, rather than collapsing into identical vectors.

Empirically, in our runs, we observe small accuracy drops (typically $< 1\text{-}2$) in a few cases, for example, Food101 (ViT-B/16), and Aircraft (RN50), while substantially lowering ECE. Overall accuracy remains comparable, and making the trade-off favorable for improved calibration. A-TPT actually improves accuracy over TPT, suggesting that angular diversity can help disambiguate fine-grained features rather than degrade them.

In principle, any text dispersion regularizer (L2, orthogonality, or angular diversity) can degrade top-1 accuracy if over-weighted on semantically overlapping classes (near-synonymous labels or fine-grained siblings). The stability of any regularizer in these scenarios is due to its formulation as an auxiliary loss. In A-TPT, angular diversity is an auxiliary term added to the standard TPT objective (entropy minimization). TPT still boosts accuracy, and angular diversity improves calibration. In Appendix A.4, limitations, we acknowledge this trade-off and fix a moderate to improve calibration while avoiding over-regularization on near-synonyms.

