# OpenReview forum: "A-TPT: Angular Diversity Calibration Properties for Test-Time Prompt Tuning of Vision-Language Models"
_ICLR.cc/2026/Conference — ICLR 2026 Poster_

### Official Review · Reviewer_5gVD · 2025-10-25

**Soundness:** 3
**Presentation:** 2
**Contribution:** 2
**Rating:** 4
**Confidence:** 4

**Summary:**

normalized textual features on the unit hypersphere, ensuring uniform distribution an A-TPT enhances test-time prompt tuning (TPT) by introducing angular diversity calibration. Instead of maximizing dispersion or orthogonality alone, it maximizes the minimum pairwise angular distance between d better

**Strengths:**

1.	Elegant mathematical framing grounded in Tammes best-packing problem.
2.	Clearly articulated motivation: poor calibration under low angular diversity.
3.	Demonstrates lower Expected Calibration Error (ECE) with negligible accuracy loss.
4.	Theoretically principled and easy to integrate.

**Weaknesses:**

1.	Incremental relative to O-TPT and C-TPT; lacks major conceptual leap.
2.	No runtime or convergence analysis for numerical optimization.
3.	Experiments mostly on classification; unclear utility for generative tasks.

**Questions:**

1.	How sensitive is A-TPT to initialization of textual prompts?
2.	Can angular diversity loss degrade accuracy for semantically overlapping classes?
3.	Is there any theoretical bound on ECE improvement from uniform angular spacing?

---

> ### Author Response · Authors · 2025-11-21
> **Rebuttal by Authors**
>
> We thank Reviewer 5gVD for summarizing the contribution of our paper and for recognizing the novelty of the paper. We appreciate his insightful and constructive comments. Below, we respond to the comments point by point.
>
> # [Response to Weaknesses]
>
> ### **Weakness 1**
>
> We appreciate the comments from the reviewer. We respectfully clarify that our work is not incremental relative to O-TPT and C-TPT.
>
> Firstly, the difference between ours and previous research (C-TPT [1], O-TPT [2]) is discussed in Secs. 1 and 3, which show that increased text dispersion improves calibration. L2 distance (C-TPT) disperses textual features away from their centroid, and orthogonality constraints (O-TPT) pursue orthogonality for all the pairwise textual features, which does not always guarantee angular separation (Sec. 3 analysis L282-314). As discussed in Sec. 1, L2 regularization can still result in textual features lying closely together, and orthogonal regularization tends to group textual features closer, when the number of classes is greater than the embedding dimension ($N>|D|$), while underutilizing hyperspherical space when $N<|D|$ (Fig. 2).
>
> Secondly, we introduce angular diversity (A-TPT), which directly maximizes the minimum angular distance on the unit hypersphere and focuses on the uniform distribution. (Sec. 3, Angular Diversity L315-340, Eqs. 1-2). Fig. 4 and its analysis show O-TPT’s inconsistency and A-TPT’s stable, uniform angular separation. Our results and analyses have more scope than both C-TPT and O-TPT. Unlike them, which suffer from poor calibration, we directly optimize the angular distance, which targets the $N>|D|$, and $\theta\rightarrow0$ (avoid stuck in near-colinear) regime that hurts calibration.
>
> Thirdly, inspired by insights from ArcFace [3], Fig. 2, 4, and Appendix A.16 show that angular diversity achieves the greatest possible minimum pairwise angular distance and explain why hyperspheres offer optimal feature separation for both $N < |D|$ and $N > |D|$ cases, yielding the most diverse prompt vectors.
>
> Fourthly, we argue that the improvement over other methods is significant. For example, in Tab. 1, cases with $N > |D|$ tend to show elevated ECE, and in these challenging points, angular diversity (A-TPT) reduces the average ECE from 4.27 (O-TPT) to 2.92, and from 4.44 to 3.60 for $N < |D|$ with the CLIP ViT-B/16 backbone. Across fine-grained datasets, A-TPT consistently outperforms C-TPT/O-TPT for both CLIP ViT-B/16 (from 5.13/4.23 (C-TPT/O-TPT) to 2.61) and RN50 (from 6.19/5.45 to 2.92) (Tab. 2). In Fig. 5, reliability diagrams show how A-TPT mitigates over- and under-confidence. Under natural distribution shifts, A-TPT further lowers average ECE, CLIP ViT-B/16 (from 5.82/4.88 (C-TPT/O-TPT) to 3.92), RN50 (from 12.1/9.69 to 7.82). A-TPT also generalizes well to medical datasets.
>
> In Appendix A.6 (Eq. 4-5, Fig. 7), the gradient analysis of our angular diversity (A-TPT) shows stable gradients that don’t vanish as $\theta\rightarrow0$ (avoid getting stuck in near-collinear) regime, whereas O-TPT suffers from poor calibration as the gradient vanishes.
>
> Based on the Tammes problem, the uniform distribution (best packing) means that the distance between two points is maximized. Therefore, while A-TPT builds upon the TPT framework, it introduces a novel angular diversity that is optimized to the utmost extent. Angular diversity aims to maximize the angular distance between textual features corresponding to their actual class labels, thereby maximizing the inter-class feature separability for better prompt calibration.
>
> Prior work [4] has shown that uniformity (uniformly distributed feature points on the unit hypersphere) preserves maximal information, closely associated with strong zero-shot CLIP performance. Therefore, we argue that uniformly distributed text features are the way to ensure better class separation and improve VLM calibration. (Sec. 1 Tammes/uniformity background & Sec. 3 motivation for angular diversity).
>
> ### **Weakness 2**
>
> We completely agree with and understand the importance of runtime or convergence analysis for numerical optimization.
>
> In Appendix A.7, Tab. 8, we have provided runtime analysis comparing the proposed numerical optimization (A-TPT) with ATFD optimization (C-TPT) and Angular optimization (O-TPT). We report (i) asymptotic complexity, (ii) Time/batch (s), and (iii) memory usage.
>
> Numerical optimization (A-TPT’s angular diversity) slightly increases the asymptotic complexity, calculation time, and memory usage compared to ATFD optimization (C-TPT’s L2), rising from 1.055s and 21838MiB to 1.058s and 21840MiB, while effectively reducing ECE. In contrast, angular optimization (O-TPT’s orthogonality) greatly increases to 1.064s and 23740MiB, due to expensive pairwise matrix operations, compared to ATFD and numerical optimization. In terms of ECE, numerical optimization (angular diversity) achieves a substantially lower ECE than both ATFD and Angular optimization.

---

> ### Author Response · Authors · 2025-11-21
> **Rebuttal by Authors**
>
> In terms of ECE, numerical optimization (angular diversity) achieves a substantially lower ECE than both ATFD and Angular optimization. It acts as a simple plug-in regularizer with negligible computational overhead, producing better calibration while preserving accuracy.
>
> In Appendix A.6 (Eq. 4-5, Fig. 7), we have provided a theoretical derivation of the gradient analyses of our angular diversity (A-TPT). This analysis shows stable gradients that don’t vanish as $\theta \rightarrow 0$ (avoiding getting stuck in the near-collinear regime), whereas O-TPT suffers from poor calibration as the gradient vanishes. This explains why A-TPT converges reliably to uniformity while O-TPT is getting stuck in near-collinear regimes ($\theta \rightarrow 0$).
>
> ### **Weakness 3**
>
> We appreciate the reviewer’s interest in the broader applicability of our method. We respectfully clarify that A-TPT is explicitly designed for the Test-Time Prompt Tuning (TPT) framework, which utilizes pre-trained Vision-Language Models (VLMs) like CLIP for zero-shot discriminative classification, not generative tasks. However, the geometric principles of A-TPT are directly transferable to generative workflows involving finite candidate selection.
>
> While A-TPT does not generate tokens auto-regressively, it is defined on textual embeddings for a finite candidate set without assuming labels or a classifier head. This makes A-TPT directly applicable to generative workflows that reduce to one-of-k selection decisions, such as multiple-choice VQA (selecting the best answer from generated options), RAG (improving the separation of retrieved passage embeddings before selection), or re-ranking (selecting the best candidate from top-k decoded outputs, e.g., tool use or style choice). In these settings, we can apply the same A-TPT objective, angular diversity, which maximizes the angular distance of the candidate embeddings. This ensures that the options are maximally distinguishable on the hypersphere, improving the calibration of the selection process under distribution shifts without retraining.
>
> For experiments, we closely follow TPT/TTA work (e.g., TPT, C-TPT, O-TPT). Therefore, we focused on classification tasks because they provide standard evaluation metrics (like Acc, ECE, and SCE), and fair comparison with prior work, which also covers the same evaluation scope. Limitation: We acknowledge that fully open-ended token generation (without a discrete candidate set) is out of scope for A-TPT. We focused on the embedding geometry to ensure that prompt vectors do not collapse as $\theta \rightarrow 0$ (avoid getting stuck in near-collinear). This is crucial for any VLM discriminative task, where the distinctness of semantic features is required for reliability.
>
> While A-TPT was validated on classification to match TPT literature standards, its core mechanism, enforcing uniformity on the hypersphere for a set of vectors, is valid for any generative workflow requiring finite candidate selection.
>
> # [Response to Questions]
>
> ### **Question 1**
>
> We completely agree and understand the importance of the sensitivity of A-TPT to the initialization of textual prompts.
>
> To evaluate this, we have examined A-TPT’s calibration performance sensitivity against C-TPT, O-TPT under different initializations of textual prompts with varying numbers of tunable context tokens across CLIP ViT-B/16 and CLIP RN50 backbones, in many tables, Sec. 4, Tab. 2 & 3 ("a photo of a [class]" - 4 tunable context tokens) and Appendix A.18, Tab. 14 ("a photo of the cool [class]" - 5 tokens) & Tab, 15 ("an example of [class]" - 3 tokens).
>
> The results consistently demonstrate that A-TPT is not sensitive to prompt initialization in a way that degrades performance, rather, it acts as a robust stabilizer that consistently maintains strong calibration capabilities (reduces calibration errors) across different prompt initializations (regardless of prompt template or tunable context token length), showcasing its robustness, further supporting its adaptability in diverse settings, and affirming its potency in better prompt-based VLM calibration.
>
> ### **Question 2**
>
> We appreciate your insightful feedback regarding the degradation of accuracy for angular diversity in semantically overlapping classes.
>
> | Method | Metric | OxfordPets | Stanford Cars | Aircrafts | Flowers102 | Food101 | Overall |
> | --- | --- | ---: | ---: | ---: | ---: | ---: | ---: |
> **CLIP ViT-B/16**
> | TPT | Acc. | 87.10 | 66.30 | 23.40 | 69.00 | 84.70 | 66.10 |
> |  | ECE | 5.77 | 5.16 | 16.80 | 13.50 | 3.98 | 9.04 |
> | C-TPT | Acc. | 88.20 | 65.80 | 24.85 | 69.80 | 83.70 | 66.47 |
> |  | ECE | 1.90 | 1.59 | 4.36 | 5.04 | 3.43 | 3.26 |
> | O-TPT | Acc. | 87.95 | 64.53 | 23.64 | 70.07 | 84.13 | 66.06 |
> |  | ECE | 1.90 | 1.78 | 3.68 | 3.87 | 1.46 | 2.54 |
> | A-TPT (Ours) | Acc. | 88.33 | 65.78 | 23.76 | 69.22 | 83.64 | 66.15 |
> |  | ECE | 1.17 | 1.09 | 3.14 | 3.61 | 1.37 | **2.08** |

---

> ### Author Response · Authors · 2025-11-21
> **Rebuttal by Authors**
>
> | Method | Metric | OxfordPets | Stanford Cars | Aircrafts | Flowers102 | Food101 | Overall |
> | --- | --- | ---: | ---: | ---: | ---: | ---: | ---: |
> **CLIP RN50**
> | TPT | Acc. | 84.50 | 58.00 | 17.00 | 62.50 | 74.90 | 59.38 |
> |  | ECE | 3.65 | 3.76 | 16.10 | 13.40 | 5.25 | 8.43 |
> | C-TPT | Acc. | 84.10 | 56.50 | 17.00 | 65.20 | 74.70 | 59.50 |
> |  | ECE | 2.77 | 1.94 | 10.70 | 4.14 | 1.86 | 4.28 |
> | O-TPT | Acc. | 83.40 | 56.44 | 16.77 | 65.61 | 74.22 | 59.29 |
> |  | ECE | 3.50 | 1.69 | 8.18 | 2.50 | 1.20 | 3.41 |
> | A-TPT (Ours) | Acc. | 83.48 | 57.08 | 14.58 | 64.89 | 74.10 | 58.83 |
> |  | ECE | 2.47 | 1.38 | 6.14 | 2.39 | 1.11 | **2.70** |
>
> In Appendix A.21 Tab. 17, we distinguish A-TPT results on fine-grained, semantically overlapping datasets (where inter-class angles are naturally small) like OxfordPets (37 breeds), StanfordCars (196 models), FGVC Aircraft (100 variants), Flowers102, and Food101 across CLIP-ViT-B/16 and RN50 backbone. Our results show that A-TPT preserves the semantic structure required for high accuracy while drastically improving calibration.
>
> - CLIP ViT-B/16: Overall Acc. remains stable (66.10 $\to$ 66.15), while ECE drops significantly (9.04 $\to$ 2.08). Notably, accuracy actually improves on datasets like OxfordPets (87.10 $\to$ 88.33) and Aircrafts (23.40 $\to$ 23.76)
>
> - CLIP RN50: We observe a minor trade-off in overall accuracy (59.38 $\to$ 58.83), while ECE drops significantly (8.43 $\to$ 2.70). Notably, accuracy actually improves on datasets like Flowers102 (62.50 $\to$ 64.89),
>
> Empirically, in our runs, we observe small accuracy drops (typically $<$ 1-2$\%$) in a few cases, for example, Food101 (ViT-B/16), and Aircraft (RN50), while substantially lowering ECE. Overall accuracy remains comparable, and making the trade-off favorable for improved calibration. A-TPT actually improves accuracy over TPT, suggesting that angular diversity can help disambiguate fine-grained features rather than degrade them.
>
> Unlike orthogonality constraints (O-TPT) that force features to be $90^{\circ}$ apart, potentially breaking semantic clusters, A-TPT solves the Tammes best-packing problem. It maximizes the minimum angular distance available. This allows semantically overlapping classes to remain relatively closer to each other than to dissimilar classes, provided they utilize the hypersphere effectively, rather than collapsing into identical vectors
>
> In principle, any text dispersion regularizer (L2, orthogonality, or angular diversity) can degrade top-1 accuracy if over-weighted on semantically overlapping classes (near-synonymous labels or fine-grained siblings). The stability of any regularizer in these scenarios is due to its formulation as an auxiliary loss. In A-TPT, angular diversity is an auxiliary term added to the standard TPT objective (entropy minimization). TPT still boosts accuracy, and angular diversity improves calibration. In Appendix A.4, limitations, we acknowledge this trade-off and fix a moderate to improve calibration while avoiding over-regularization on near-synonyms.
>
> ### **Question 3**
>
> We do not claim a distribution-free theoretical bound on ECE improvement. In Sec. 1 and 3, our theory gives a geometric guarantee of uniformly spaced class-wise textual features (maximizing the minimum pairwise angular distance), and it analyzes optimization behavior (stable gradients), but it does not provide a closed-form ECE bound.
>
> Because ECE depends on the data distribution and the softmax temperature, uniform spacing only constrains geometry, not the mapping from logits to calibrated confidence. Under cosine-softmax with temperature, increasing the minimum pairwise angular distance by $\Delta\theta\_\min$ perturbs logits by at most $O(\Delta\theta\_\min)$, so predicted confidences change by at most $O(\Delta\theta\_\min/\tau)$. This is a stability inequality, not a bound on ECE improvement.
>
> We empirically show consistent ECE drops when angular diversity increases (see Tab. 1 grouped results): $N>|D|$ from 4.27 (O-TPT) to 2.92 (A-TPT) and $N<|D|$ from 4.44 to 3.60 with CLIP ViT-B/16 backbone.
>
> ```
> References
> [1] Yoon et al. C-tpt: Calibrated test-time prompt tuning for vision-language models via text feature dispersion, ICLR 2024
> [2] Sharifdeen et al. O-TPT: Orthogonality Constraints for Calibrating Test-time Prompt Tuning in Vision-Language Models, CVPR 2025
> [3] Deng et al. Arcface: Additive angular margin loss for deep face recognition, CVPR 2019
> [4] Wang et al. Understanding contrastive representation learning through alignment and uniformity on the hypersphere, ICML 2020

---

> ### Author Response · Authors · 2025-11-28
> **Request for feedback and further discussion on rebuttal**
>
> Dear Reviewer 5gVD,
>
> We truly appreciate your time and effort in reviewing our work and providing thoughtful feedback. In our rebuttal above, we have thoroughly replied to each of the concerns raised in your earlier comments.
>
> As the author–reviewer discussion phase is coming to a close, we would be grateful if you could confirm whether our responses have addressed your concerns. We would greatly appreciate any follow-up questions or clarifications you may have to engage further if needed. Your additional input will help ensure a fair and thorough assessment of our work.
>
> We sincerely hope to hear your feedback. We would also appreciate it if you could consider raising your rating of this submission, given our additional results and analysis.
>
> Sincerely,
>
> Authors of Paper #16099

---

### Official Review · Reviewer_RJNf · 2025-10-30

**Soundness:** 4
**Presentation:** 2
**Contribution:** 3
**Rating:** 6
**Confidence:** 4

**Summary:**

This paper describes a new criterion for test-time prompt tuning in order to reduce calibration error.  It has previously been noted that calibration error can be reduced by designing a prompt template so that, when the different class labels are inserted into the template, the resulting prompts have maximally dispersed text vectors.  C-TPT measured dispersion using mean L2 distance, which does not guarantee pairwise separation; O-TPT guaranteed pairwise dispersion of 90 degrees if the number of classes is less than twice the number of embedding dimensions, but not otherwise.  The proposed A-TPT minimizes the maximum pairwise cosine similarity of classes, thus maximizing pairwise dispersion.

**Strengths:**

Derivations are interesting and clear.

Equations and derivations seem correct. The point about \arccos normalizing gradient magnitudes is quite interesting.  I find multi-letter variable names aesthetically displeasing in general, but the use of "Cos" as a variable name does not impair legibility or correctness in this case.

Results show significant consistent reduction in calibration error, with small and inconsistent changes in accuracy, across 15 datasets, in comparison to TPT, C-TPT, and O-TPT.

**Weaknesses:**

Minor: Fig. 3 clearly shows that the prompts with the highest ECE ("the nearest shape in this image is" and TPT) are clustered in the center, while other prompts are distributed.  This does not show, however, that the prompts with high ECE have low angular diversity, because t-SNE does not show the angles of vectors: it only shows their cluster structure.

**Questions:**

On p. 4, what does it mean when the same prompt appears in both the list "Hard prompts" and the list "Tuned prompts," but with different Accuracy, ECE and AD?

Eq. (1) min_{j,j\ne i} should be min_{i,j\ne i}

---

> ### Author Response · Authors · 2025-11-21
> **Rebuttal by Authors**
>
> We thank Reviewer RJNf for summarizing the contribution of our paper and for recognizing the novelty of the paper. We appreciate his insightful and constructive comments. Below, we respond to the comments point by point.
>
> # [Response to Weaknesses]
>
> ### **Weakness 1**
>
> Thank you for pointing this out. In the revised paper, we have updated our notation by removing the Cos to avoid potential Cos vs \cos confusion and reducing the use of multi-letter variable names while keeping the mathematical formulation identical
>
> ### **Weakness 2**
>
> Thank you for pointing this out. We completely agree that t-SNE does not show the angles of vectors: it only shows their local neighborhood cluster structure. Our intent with Fig. 3 was to provide a qualitative illustration that prompts with high-ECE prompts tend to collapse into dense clusters, while well-calibrated prompts exhibit greater dispersion.
>
> To substantiate the claim quantitatively, that high-ECE prompts possess low angular diversity, without relying solely on t-SNE, we report angular-diversity (AD) numbers for the same prompts. We point to those in Sec. 3 directly from the Fig. 3 legends. For example, the two high-ECE prompts have notably lower AD than the better-calibrated ones. As hypothesized in Sec. 3, there is a negative correlation between ECE and AD within the same accuracy group.
>
> # [Response to Questions]
>
> ### **Question 1**
>
> Thank you for raising this point.
>
> - **Hard prompts** refer to hand-crafted templates evaluated in a zero-shot setting (without test-time prompt tuning). The metrics reported (Acc., ECE, AD) reflect the performance of the pre-trained CLIP RN50 on Caltech101 using those fixed raw strings without any test-time optimization.
>
> - **Tuned prompts** refer to the state of the prompt vectors after test-time optimization (with TPT or A-TPT). The metrics shown are after test-time prompt tuning, which is why Acc/ECE/AD change. Here, the same Hard prompt template is used as the prompt initialization. The list even labels which method was used (e.g., TPT, TPT + A-TPT).
>
> The Acc., ECE, and AD values differ because the "Hard" metrics reflect the static initialization, whereas the "Tuned" metrics reflect the adapted feature space.
>
> That section was set up to show the negative correlation between ECE and AD within the same accuracy groups across 80 prompt styles.
>
> ### **Question 2**
>
> Thank you for pointing out this typo.
>
> Eq. (1) should read $\min_{j \in \{\{1, ..., N\} \textbackslash \{i\}}}$
>
> The index $i$ is handled by the outer average, the inner minimization is only over $j$ with $j \neq i$. Writing $\min_{i,j \neq i}$ would change the objective to target the single global worst-case pair, which is not what we optimize (and would not match the code that takes a row-wise max cosine to compute the nearest neighbor per class $i$ individually)
>
> We corrected the notation in the revised paper and use $\min_{j \in \{1,\dots,N\} \setminus \{i\}}$ (minimizing over $j$ such that $j \neq i$ for a fixed $i$) to avoid any ambiguity while accurately reflecting the row-wise optimization logic.

---

> ### Author Response · Authors · 2025-11-28
> **Request for feedback and further discussion on rebuttal**
>
> Dear Reviewer RJNf,
>
> We truly appreciate your time and effort in reviewing our work and providing thoughtful feedback. In our rebuttal above, we have thoroughly replied to each of the concerns raised in your earlier comments.
>
> As the author–reviewer discussion phase is coming to a close, we would be grateful if you could confirm whether our responses have addressed your concerns. We would greatly appreciate any follow-up questions or clarifications you may have to engage further if needed. Your additional input will help ensure a fair and thorough assessment of our work.
>
> We sincerely hope to hear your feedback. We would also appreciate it if you could consider raising your rating of this submission, given our additional results and analysis.
>
> Sincerely,
>
> Authors of Paper #16099

---

### Official Review · Reviewer_r5c7 · 2025-11-01

**Soundness:** 3
**Presentation:** 3
**Contribution:** 2
**Rating:** 4
**Confidence:** 4

**Summary:**

This paper targets the calibration problems that arise when doing test-time prompt tuning (TPT) on vision-language models and argues that existing fixes (like text feature dispersion (C-TPT) and orthogonality constraints (O-TPT)) don’t actually guarantee that class-wise text features are well separated, especially when the number of classes exceeds the embedding dimension. To address this, the authors propose A-TPT (Angular Test-time Prompt Tuning), which adds an angular diversity regularizer that, for each class embedding, maximizes its minimum angular distance to any other class, encouraging a more uniform packing of text features on the unit hypersphere. According to experiments on fine-grained, distribution-shifted the proposed method reduces expected calibration error (ECE) while largely preserving TPT’s accuracy gains over zero-shot CLIP and prior TPT variants.

**Strengths:**

* The paper clearly identifies the shortcomings of prior text-feature dispersion approaches and uses Figure 2 to illustrate them effectively.

* The reported ECE gains over baselines such as C-TPT and O-TPT are also encouraging.

* The authors show the method also works not only on standard benchmarks used to evaluate CLIP performance, but also on 'calibration critical applications' such as medical domain in Table 4.

**Weaknesses:**

* Although the paper proposes angular diversity regularization as a new metric, the method still operates within the existing C-TPT and O-TPT test-time adaptation paradigm, so the contribution feels more incremental than fundamentally novel in terms of theory or technique

* Could the proposed method be a complementary to previous methods (e.g., C-TPT or O-TPT). That is, could we for example enforce the proposed angular diversity on top of textual dispersion proposed by C-TPT or the orthogonality constraints of O-TPT. It would be interesting to see such an ablation.

* Since the ECE metric could suffer from bias, could the authors report calibration metrics other than ECE as well?

**Questions:**

See weaknesses above.

---

> ### Author Response · Authors · 2025-11-21
> **Rebuttal by Authors**
>
> We thank Reviewer r5c7 for summarizing the contribution of our paper and for recognizing the novelty of the paper. We appreciate his insightful and constructive comments. Below, we respond to the comments point by point.
>
> # [Response to Weaknesses & Questions]
>
> ### **Weakness 1**
>
> We respectfully clarify that the motivation and positioning behind this are explained in Secs. 1 and 3. The pursuit of increased classification accuracy during test time necessitates incorporating entropy minimization (TENT [1]), which leads to overconfident predictions [2] and, unfortunately, worsens the calibration error. In practical deployments, it has substantial significance because miscalibration in models deployed in safety-critical applications can be harmful in two ways: overconfidence often results from models assigning high confidence to incorrect predictions, and underconfidence is associated with uncertain or unreliable confidence estimation even if the model makes correct predictions. From a scientific perspective, it poses questions regarding model reliability and the limitations of current calibration techniques (See Fig. 5 for over- and under-confidence of C-TPT and O-TPT). Therefore, improving VLM calibration is often overlooked, and addressing it is a worthwhile and pressing problem in the TPT.
>
> Firstly, the difference between ours and previous research (C-TPT [3], O-TPT [4]) is discussed in Secs. 1 and 3, which claim that the lack of text dispersion can hurt the calibration performance. L2 distance (C-TPT) disperses textual features away from their centroid, and orthogonality constraints pursue orthogonality for all the pairwise textual features (O-TPT), which do not guarantee angular separation (Sec. 3 analysis L282-314). As discussed in Sec. 1, L2 regularization can still result in textual features lying closely together. Furthermore, orthogonal regularization tends to group textual features closer, when the number of classes is greater than the embedding dimension ($N > |D|$), while underutilizing hyperspherical space when $N < |D|$ (Fig. 2).
>
> Secondly, we propose a simple yet powerful method, angular diversity (A-TPT), to solve this issue, which directly maximizes the minimum angular distance on the unit hypersphere (Eqs. 1-2) and focuses on the uniform distribution (Eqs. 1-2). (Sec. 3, Angular Diversity L315-340). Our proposed method seeks to set a balance, aiming to reduce this calibration degradation while preserving accuracy benefits. Fig. 4 and its analysis show O-TPT’s inconsistency and A-TPT’s stable, uniform angular separation. Our results and analyses have more scope than both C-TPT and O-TPT. Unlike them, which suffer from poor calibration, we directly optimize the angular distance. This targets the $N > |D|$, and $\theta \to 0$ (avoiding getting stuck in near-collinear) regime that hurts calibration.
>
> Thirdly, inspired by insights from ArcFace [3], Figures 2, 4, and Appendix A.16 show that angular diversity achieves the greatest possible minimum pairwise angular distance and discuss why hyperspheres offer optimal feature separation for both $N < |D|$ and $N > |D|$ cases, yielding the most diverse prompt vectors.
>
> Fourthly, we argue that the improvement over other methods is significant. For example, in Tab. 1, cases with $N > |D|$ tend to show elevated ECE, and in these challenging points, angular diversity (A-TPT) reduces the average ECE from 4.27 (O-TPT) to 2.92, and from 4.44 to 3.60 for N<∣D∣ with the CLIP ViT-B/16 backbone. Across fine-grained datasets, A-TPT consistently outperforms C-TPT/O-TPT for both CLIP ViT-B/16 (reducing ECE from 5.13/4.23 to 2.61) and RN50 (from 6.19/5.45 to 2.92) (Tab. 2). In Fig. 5, reliability diagrams show how A-TPT mitigates over- and under-confidence. Under natural distribution shifts, A-TPT further lowers average ECE: CLIP ViT-B/16 from 5.82/4.88 (C-TPT/O-TPT) to 3.92, RN50 from 12.1/9.69 to 7.82. A-TPT also generalizes well to medical datasets.
>
> In Appendix A.6 (Eq. 4-5, Fig. 7), the gradient analysis of our angular diversity (A-TPT) shows stable gradients that don’t vanish as $\theta \to 0$ (avoid getting stuck in near-colinear) regime, whereas O-TPT suffers from poor calibration as the gradient vanishes.
>
> Based on the Tammes problem, the uniform distribution (best packing) means that the distance between two points is maximized. Therefore, while A-TPT builds upon the TPT framework, it introduces a novel angular diversity that is optimized to the utmost extent. Angular diversity aims to maximize the angular distance between textual features corresponding to their actual class labels, thereby maximizing the inter-class feature separability for better prompt calibration.

---

> ### Author Response · Authors · 2025-11-21
> **Rebuttal by Authors**
>
> Prior work [5] has shown that uniformity (uniformly distributed feature points on the unit hypersphere) preserves maximal information, closely associated with strong zero-shot CLIP performance, and as a plug-in regularizer without changing the pre-trained backbone, the only effect of angular diversity, maximizing the angular distance, is to promote the uniformity of textual features. Therefore, we argue that uniformly distributed text features are the way to ensure better class separation and improve VLM calibration. (Sec. 1 Tammes/uniformity background & Sec. 3 motivation for angular diversity).
>
> ### **Weakness 2**
>
> We completely agree with and understand the importance of ablations to see whether the proposed method is complementary to previous methods (e.g., C-TPT or O-TPT).
>
> In Appendix A.19, Tab. 16, we have presented the calibration results of a combined A-TPT and C-TPT, which yields lower ECE than either method individually across DTD/Flowers102/UCF101 datasets, lowering ECE from 11.90/5.04/2.54 (C-TPT) and  4.76/3.61/2.12 (A-TPT) to 4.11/3.49/1.96 (A-TPT + C-TPT).
>
> This ablation aims to enforce the proposed angular diversity, maximizing the minimum angular distance (A-TPT) on top of textual dispersion, maximizing the L2-distance proposed in C-TPT. The findings indicate that combining A-TPT with C-TPT could be complementary to C-TPT, leading to superior calibration performance that outperforms A-TPT alone.
>
> ### **Weakness 3**
>
> We completely agree and understand the importance of report calibration metrics other than ECE, since the ECE metric could suffer from bias.
>
> In Appendix A.14, we have reported calibration results using the SCE metric. SCE serves as a class-wise variant of ECE, which calculates calibration error for each class separately before averaging, avoiding the cancellation of over-confidence and under-confidence across different classes within the same bin. Tab. 13 compares the SCE results across ten fine-grained classification datasets with the CLIP ViT-B/16 and CLIP RN50 backbones. A-TPT consistently achieves superior calibration performance against Baseline, TPT, C-TPT, and O-TPT, demonstrating a substantial reduction in SCE across all datasets.
>
> These results confirm that the calibration improvements of A-TPT are robust and not an artifact of ECE binning bias.
>
> ```
> References
> [1] Wang et al. Tent: Fully test-time adaptation by entropy minimization, ICLR 2021
> [2] Mukhoti et al. Calibrating deep neural networks using focal loss, NeurIPS 2020
> [3] Yoon et al. C-tpt: Calibrated test-time prompt tuning for vision-language models via text feature dispersion, ICLR 2024
> [4] Sharifdeen et al. O-TPT: Orthogonality Constraints for Calibrating Test-time Prompt Tuning in Vision-Language Models, CVPR 2025
> [5] Wang et al. Understanding contrastive representation learning through alignment and uniformity on the hypersphere, ICML 2020

---

> ### Author Response · Authors · 2025-11-28
> **Request for feedback and further discussion on rebuttal**
>
> Dear Reviewer r5c7,
>
> We truly appreciate your time and effort in reviewing our work and providing thoughtful feedback. In our rebuttal above, we have thoroughly replied to each of the concerns raised in your earlier comments.
>
> As the author–reviewer discussion phase is coming to a close, we would be grateful if you could confirm whether our responses have addressed your concerns. We would greatly appreciate any follow-up questions or clarifications you may have to engage further if needed. Your additional input will help ensure a fair and thorough assessment of our work.
>
> We sincerely hope to hear your feedback. We would also appreciate it if you could consider raising your rating of this submission, given our additional results and analysis.
>
> Sincerely,
>
> Authors of Paper #16099

---

### Official Review · Reviewer_Aoux · 2025-11-01

**Soundness:** 3
**Presentation:** 3
**Contribution:** 2
**Rating:** 6
**Confidence:** 4

**Summary:**

The paper proposes Angular Diversity for Test-time Prompt Tuning (A-TPT), instead of pushing text prompts to be dispersed by l2 distance or cosine similarity, it maximizes the minimum pairwise angle between normalized class-wise prompt vectors (a maximin objective using θ=arccos of cosine similarity). This directly spreads prompts on the unit hypersphere to promote more uniform coverage and better calibration during inference. The authors also argue why this angle-based objective has stable gradients even when vectors are very close—unlike the orthogonality loss whose gradient vanishes as angles go to 0.

**Strengths:**

- Rather than optimizing L2 distance or cosine similarity, the paper optimizes the **angle itself,** which better captures geometric separation on the unit sphere and compensates for the shortcomings of previous work. This paper shows the limitation of previous work well.
- The paper includes extensive analyses that illuminate the method’s behavior from multiple perspectives, aiding interpretation and practical use.
- It explicitly examines the calibration differences between N > |D| and N ≤ |D|, a case prior work largely overlooks, and clarifies where the proposed method offers the biggest gains over O-TPT.

**Weaknesses:**

- When we increase λ, we understand this as trading some accuracy for improved ECE (better calibration). This trend aligns with Flowers102, but Food101 shows a contrasting pattern. Could you provide insight into why the two datasets behave differently? Also, are these curves averaged over multiple seeds, and how large is the variance across runs?
- How did you choose the λ term?
- In the main performance table, could you report results separately or make them explicitly distinguishabl for the N>|D| and N≤|D| regimes? This would help isolate where your method provides the most benefit over O-TPT.
- How do you ensure numerical stability when computing arccos?
- While increasing the minimun θ can encourage dispersion, it doesn’t seem sufficient to prevent localized density (clustering) in certain regions. Do you have any guarantees or empirical evidence that your method avoids such partial clustering?

**Questions:**

See weakness section

---

> ### Author Response · Authors · 2025-11-21
> **Rebuttal by Authors**
>
> We thank Reviewer Aoux for summarizing the contribution of our paper and for recognizing the novelty of the paper. We appreciate his insightful and constructive comments. Below, we respond to the comments point by point.
>
> # [Response to Weaknesses & Questions]
>
> ### **Weakness 1**
>
> Thank you for raising this point. The difference in calibration behaviour of the Flowers102 and Food101 datasets when $\lambda$ increases is due to dataset characteristics and how model confidence (over- vs. under-confidence) is distributed.
>
> Refer to Fig. 5, the reliability diagrams of C-TPT and O-TPT show overconfidence on Flowers102 and underconfidence on Food101.
>
> - Flowers102: likely contains more fine-grained, visually similar classes, where the baseline model often makes incorrect predictions with high confidence (overconfidence). To correct overconfidence, the model must lower its confidence in ambiguous samples. This necessary softening of the probability distribution can occasionally shift a high-confidence incorrect prediction to a low-confidence correct or incorrect state. This results in the observed trade-off with increased $\lambda$, trading some accuracy as the model becomes more conservative to improve ECE.
>
> - Food101: features more visually distinct classes, where the model makes correct predictions with uncertain or unreliable confidence estimates (under-confidence). Increasing $\lambda$ maximizes the margin between these distinct features. This pushes the correct class probabilities higher (mitigating under-confidence) without introducing ambiguity. This helps A-TPT to lower ECE without sacrificing accuracy (or even gaining a little by sharpening the decision boundary).
>
> That’s why A-TPT shows a trend trading some accuracy for improved ECE on Flowers102, while on Food101, A-TPT shows a contrasting pattern, lower ECE without sacrificing accuracy (or even gaining a little) for better calibration.
>
> As shown in the Pareto frontier analysis (Appendix A.20, Fig. 14), A-TPT seeks a better balance overall rather than sacrificing accuracy for better calibration, exactly as observed on Food101, while the accuracy-ECE trade-off on Flower102 across $\lambda$, confirms that increased performance could not be a result of a better $\lambda$.
>
> The reported curves are averaged over three seed runs. In Appendix A.13 Tab. 12, we have reported 3-seed variability for A-TPT (ViT-B/16), with low standard deviations (Flowers102: Std Acc. $\pm 0.09$, Std ECE $\pm 0.15$; Food101: Std Acc. $\pm 0.04$, Std ECE $\pm 0.10$), confirming the curve shapes are stable across runs.
>
> ### **Weakness 2**
>
> We performed an ablation by sweeping $\lambda$ to select a fixed value (without per-dataset tuning) for all runs to keep the test-time setup label-free. In Appendix A.10, implementation details, we use $\lambda$ of 80 for all main experiments. For the natural-distribution-shift datasets, we set $\lambda$ to 10.
>
> Since we cannot tune $\lambda$ on a labeled held-out set with our test-time adaptation setting (making hyperparameter tuning difficult) (see Appendix A.4, limitations), we selected a single fixed $\lambda$ from the Pareto frontier analysis by sweeping $\lambda$ that balances accuracy and calibration.
>
> In Appendix A.20, Fig. 14, the Pareto frontier curves show the accuracy-ECE trade-off across $\lambda$, confirming that increased performance could not be a result of a better $\lambda$. In Appendix A.4 limitations, we acknowledge that a fixed $\lambda$ may not be optimal for every sample, exploring adaptive $\lambda$ as future work.
>
> ### **Weakness 3**
>
> We completely agree with and understand the importance of explicitly distinguishing results for the $N > |D|$ and $N < |D|$ regimes. In Tab. 1, we have presented the accuracy and ECE results for CLIP ViT-B/16 backbone, dividing data into two groups based on the number of classes relative to the embedding dimension of TPT text features. Group 1 includes cases with $N > |D|$, while Group 2 includes $N < |D|$. We then calculate the ECE and accuracy separately for each group, allowing a more fine-grained analysis of each method’s performance. As hypothesized, cases with $N > |D|$ tend to show elevated ECE, indicating poor calibration and suggesting these are more challenging points. In these challenging cases (Group 1), our method significantly outperforms C-TPT as well as O-TPT in terms of calibration performance, resulting in an overall lower ECE. A-TPT reduces ECE from 4.27 (O-TPT) to 2.92 for $N > |D|$, and from 4.44 to 3.60 for $N < |D|$. Notably, A-TPT provides the most benefit over O-TPT in the $N > |D|$ regime.
>
> In addition, in Appendix A.11, to ensure this trend holds with varying test set size, we provided a weighted average for the accuracy and ECE with CLIP ViT-B/16 and CLIP RN50 backbones for fair comparison.

---

> ### Author Response · Authors · 2025-11-21
> **Rebuttal by Authors**
>
> ### **Weakness 4**
>
> Thank you for raising this point. The arccos operation suffers from gradient explosion if its input leaks outside the boundaries of $[-1, 1]$ and gets too close to the ends (as its gradient is $-1/\sqrt{1-x^2}$). In Eq. 1, where $\theta$ is the $\arccos$ of pairwise cosine similarities (dot products of normalized text features $EE^T$), we clamp within $(-1,1)$, specifically from -0.99999 to 0.99999, to prevent NaNs in the $\arccos$ calculation, and to prevent the infinite gradient at $\pm 1$ during backpropagation.
>
> In Sec. 3 and Appendix A.6, we discuss the gradient analyses and explain why directly optimizing angular distance is stable compared to cosine-based penalties in the $\theta \to 0$ (avoid getting stuck in near-collinear) regime.
>
> ### **Weakness 5**
>
> Thank you for raising this point. We don’t claim a global "perfectly uniform" guarantee, but A-TPT’s objective gives an optimal  worst-case ($N > |D|$ and $\theta \to 0$ (avoid getting stuck in near-colinear) regime) angular separation guarantee
>
> We optimize a max-min angle objective: over unit-normalized class-wise text features (Eq. 1). At any optimum, every class vector is at least some $\theta_{\min}$ away from its nearest neighbor. That lower-bounds pairwise proximity and directly targets the would-be cluster cores. This is motivated by hyperspherical best-packing (Tammes problem): maximizing the minimum angular distance promotes uniform use of the hyperspherical space, though it’s not a proof of perfect uniformity.
>
> In Sec. 3 and Appendix A.6 Fig.7, gradients come from the tightest pairs, so any nascent cluster with very small angles ($\theta \to 0$) gets pushed apart first. Unlike orthogonality losses, whose gradients vanish as $\theta \to 0$, our gradient stays stable at small angles. In Fig. 3, as $\lambda$ increases, features spread and align with class labels, poorly calibrated prompts that clustered tightly disperse with A-TPT. Fig. 4, across both regimes ($N > |D|$ and $N < |D|$), A-TPT yields more consistent cosine similarities and the greatest possible minimum angular separation per sample. O-TPT shows clustered behavior, especially when $N > |D|$. In Sec. 4 Tab. 1, A-TPT lowers ECE the most where clustering pressure is highest ($N > |D|$), consistent with better dispersion.
>
> The combination of stable gradients and the max-min objective and the analyses above indicates that A-TPT breaks partial clustering that hurts calibration, especially in $N > |D|$ and $\theta \to 0$ settings.

---

> ### Author Response · Authors · 2025-11-28
> **Request for feedback and further discussion on rebuttal**
>
> Dear Reviewer Aoux,
>
> We truly appreciate your time and effort in reviewing our work and providing thoughtful feedback. In our rebuttal above, we have thoroughly replied to each of the concerns raised in your earlier comments.
>
> As the author–reviewer discussion phase is coming to a close, we would be grateful if you could confirm whether our responses have addressed your concerns. We would greatly appreciate any follow-up questions or clarifications you may have to engage further if needed. Your additional input will help ensure a fair and thorough assessment of our work.
>
> We sincerely hope to hear your feedback. We would also appreciate it if you could consider raising your rating of this submission, given our additional results and analysis.
>
> Sincerely,
>
> Authors of Paper #16099

---

### Author Response · Authors · 2025-11-21
**General Response**

Thank you for providing constructive insights on our paper and for taking the time to review the paper. We have revised the manuscript as outlined below to incorporate reviewers' comments, with the revised manuscript written in **blue**.

- We have added distinguished A-TPT results for fine-grained, semantically overlapping datasets in Appendix A.21.
- We have highlighted relevant rebuttal sections, tables, and figures in blue.

Additional changes will be incorporated in a subsequent revision.

---

### Author Response · Authors · 2025-12-03
**Official Comment by Authors**

Dear Area Chairs,

We would like to sincerely thank the reviewers (R1: 5gVD, R2: r5c7, R3: Aoux, R4: RJNf) for their detailed review and constructive feedback and for recognizing the novelty of our approach to improving VLM calibration. We are encouraged that reviewers found the mathematical framing "elegant" and "grounded" [R1], with "interesting and clear" derivations [R4], and appreciated the "extensive analyses" and "substantial ECE gains" [R2, R3]. We have carefully addressed each noted point separately, incorporated their suggestions into the revised manuscript, and we sincerely hope our responses have clarified their concerns. We summarize four common themes raised by multiple reviewers below, followed by specific responses to each reviewer.

- **[R1, R2, R3] Novelty and comparison:** Some reviewers questioned whether A-TPT is incremental relative to C-TPT and O-TPT; it lacks a major conceptual leap.

  - **[R1, R2] Angular guarantee:** We have clarified that while previous research (C-TPT (L2 distance) and O-TPT (orthogonality constraints)) focused on improving VLM calibration via text dispersion, they fail to guarantee optimal angular separation when the number of classes exceeds the embedding dimension ($N > |D|$), while underutilizing hyperspherical space when $N<|D|$. Inspired by insights from prior work on uniformity, Tammes problem (best-packing), and ArcFace, our method, A-TPT, focuses on angular diversity, maximizing the angular distance to achieve uniformity (good class separation) and sharpening class boundaries to improve calibration performance in both $N<|D|$ and $N>|D|$ regimes.

  - **[R3] Grouped analyses:** To further demonstrate this, we explicitly distinguished results for $N<|D|$ and $N>|D|$ (Tab. 1),  showing that A-TPT provides the most significant ECE gains in the challenging $N>|D|$ regime where O-TPT fails.

  - **[R2] Complementary nature:** In Appendix A.19, we conducted an ablation study showing that A-TPT is complementary to C-TPT; combining angular diversity with text dispersion yields superior calibration, outperforming either method individually.

  - **[R1, R2] Gradient stability & convergence analyses:** In Appendix A.6 (Eq. 4-5, Fig. 7), we provided a theoretical derivation showing that our angular diversity (A-TPT) exhibits a stable gradients that don’t vanish as $\theta\to0$ (avoid getting stuck in near-collinear) regime, whereas O-TPT suffers from poor calibration as the gradient vanishes at $\theta\to0$. This explains why A-TPT converges reliably to uniformity while O-TPT is getting stuck in high-density ($N>|D|$ and $\theta\to0$) regimes.

- **[R1, R3] Runtime, convergence analyses and stability:** Reviewers asked for clarification regarding computational cost, convergence analysis, and numerical stability.

  - **[R1] Runtime analyses:** We added a runtime analyses (Appendix A.7, Tab. 8). A-TPT incurs negligible computational overhead (rising only from 1.055s and 21838MiB to 1.058s and 21840MiB per batch compared to C-TPT) while achieving substantially lower ECE, whereas O-TPT is significantly more expensive due to pairwise matrix operations.

  - **[R1] Convergence analyses:** *See above Gradient stability & convergence analyses.*

  - **[R3] Numerical stability:** gradient explosion in the arccos operation, we clarified that we clamp input values within $(-1, 1)$ to prevent NaNs and infinite gradients at $\pm 1$.

- **[R1, R3] Trade-off:**

  - **[R1] Degrade accuracy:** Reviewers asked if maximizing angular diversity degrades accuracy for fine-grained, semantically overlapping classes. In Appendix A.21, we distinguish A-TPT results on fine-grained, semantically overlapping datasets (e.g., OxfordPets, FGVC Aircraft) showing that A-TPT preserves or often improves accuracy, while substantially lowering ECE over TPT by better disambiguating features. In Appendix A.10, we selected a moderate regularization strength ($\lambda$) that prevents over-regularization on near-synonyms. We clarified that, unlike O-TPT, which strictly forces orthogonality ($90^\circ$), A-TPT maximizes the minimum available angular distance. This allows semantically similar classes to remain relatively closer to each other than to dissimilar classes, maintaining necessary semantic clustering while utilizing the hyperspherical space effectively.

  - **[R3] Pareto front analysis:** We provided a Pareto frontier analysis in Appendix A.20 (Fig. 14) to visualize the accuracy-ECE trade-off across different $\lambda$ values. We clarified the different calibration behavior (contrasting pattern) observed in datasets (e.g., Flowers102 vs. Food101), as a necessary correction of over-confidence and under-confidence, with reliability diagrams.

---

> ### Author Response · Authors · 2025-12-03
> **Official Comment by Authors**
>
> - **[R1, R2, R3] Robustness and applicability:** Reviewers raised questions regarding the method's applicability beyond classification, reliance on the ECE metric, sensitivity to textual prompts, and hyperparameter $\lambda$.
>
>   - **[R1] Utility for generative tasks:** We clarified that while A-TPT targets discriminative tasks on TPT, its geometric principles, enforcing uniformity on the hypersphere, are directly transferable to generative workflows (like enforcing uniformity on a finite set of embeddings) involving "choose-one-of-k" selection (e.g., multiple-choice VQA, RAG, re-ranking).
>
>   - **[R2] ECE metric bias:** We acknowledge that ECE can suffer from binning bias. In response, we reported results using the Static Calibration Error (SCE) metric (Appendix A.14), which calculates error class-wise. A-TPT consistently reduces SCE across all datasets, confirming that our calibration improvements are robust and not an artifact of binning bias.
>
>   - **[R1] Prompt initialization sensitivity:** Tabs. 2, 3, 14, 15 show that A-TPT is robust to prompt initialization, consistently improving calibration across various templates and tunable context token lengths.
>
>   - **[R3] Hyperparameter $\lambda$:** We selected a regularization strength ($\lambda$) that minimizes accuracy-ECE trade-off from Pareto frontier analysis in Appendix A.20 (Fig.14) by sweeping $\lambda$ values.
>
> **[R4] Clarity:** We corrected the notation in Eq. 1, simplified multi-letter variable naming, and clarified the difference between "Hard" and "Tuned" prompts to better illustrate the negative correlation between Angular Diversity and ECE.
>
> We believe that we have resolved all important concerns raised by reviewers and hope that these improvements will be sufficient for a positive evaluation.
>
> Sincerely,
>
> Authors of Paper #16099

---

### Meta-Review · Area_Chair_hBLC · 2025-12-22

**Summary:**

The paper received four reviews with mixed initial scores: 2x 4 and 2x 6.

Reviewer 5gVD (score: 4) raised three concerns: incremental contribution compared with O-TPT and C-TPT, lack of runtime/convergence analysis, and classification-only results. In terms of comparison with O-TPT and C-TPT, the rebuttal provided a detailed comparison: O-TPT and C-TPT are based on orthogonality constraint and L2 distance, respectively, whereas the proposed A-TPT is based on angular diversity. The calibration results clearly show that angular diversity significantly outperforms the two previous designs and therefore the effectiveness of A-TPT is well justified. The rebuttal also provided additional analysis on runtime and discussed convergence of the method. Overall, the AC finds the rebuttal clear and solid, addressing all concerns raised from this reviewer. The AC predicts that the reviewer should raise the score to at least 6.

Reviewer RJNf (score: 6) seems to be satisfactory with the work and only mentioned some minor weakness related to figures and notations. The rebuttal addressed these issues so the AC believes this reviewer would keep the score of 6 or raise it to 8.

Reviewer r5c7 (score: 4) questioned the novelty as the test-time adaptation paradigm remains the same for the method and baselines. This reviewer also asked whether A-TPT can be complementary to C-TPT/O-TPT and requested extra metrics other than ECE. Regarding novelty, the rebuttal reiterated the motivation of angular diversity and emphasized the clear improvement over the baselines. The AC believes the rebuttal is clear and solid: the improvement over C-TPT & O-TPT clearly demonstrates the significance of the angular diversity idea. This concern should have been cleared. In addition, the rebuttal combined A-TPT with C-TPT and as a result ECE is reduced; the rebuttal also reported results on the SCE metric in the appendix. The AC believes the rebuttal has addressed all concerns from reviewer r5c7. The score should be increased from 4 to 6 or 8.

Reviewer Aoux (score: 6) did not question the novelty nor the results. The reviewer mainly requested more insights/discussions and clarifications on some implementation details. The rebuttal has addressed these concerns. The AC believes the reviewer would at least keep the score of 6.

In summary, the most critical concern was about technical novelty. The rebuttal explained in detail the differences between A-TPT and previous TPT methods. The response was clear and solid. The AC believes that all concerns have been addressed in the rebuttal and predicts that the final scores would be 6, 6/8, 6/8, and 6.

**Reviewer Concerns:**

As discussed above, all concerns should have been addressed in the rebuttal. There is no lingering concern.

**Reviewer Scores:**

The AC believes the rebuttal is clear and solid. The negative scores would be raised to 6 or 8, while the positive scores would be maintained or raised. The final predicted scores are 6, 6/8, 6/8, and 6. The AC recommends acceptance.

---

### Decision · Program_Chairs · 2026-01-26

Accept (Poster)